# PRIORITIZED GENERATIVE REPLAY

**Renhao Wang, Kevin Frans, Pieter Abbeel, Sergey Levine, and Alexei A. Efros**
Department of Electrical Engineering and Computer Science
University of California, Berkeley

## ABSTRACT

Sample-efficient online reinforcement learning often uses replay buffers to store experience for reuse when updating the value function. However, uniform replay is inefficient, since certain classes of transitions can be more relevant to learning. While prioritization of more useful samples is helpful, this strategy can also lead to overfitting, as useful samples are likely to be more rare. In this work, we instead propose a prioritized, parametric version of an agent's memory, using generative models to capture online experience. This paradigm enables (1) *densification* of past experience, with new generations that benefit from the generative model's generalization capacity and (2) *guidance* via a family of "relevance functions" that push these generations towards more useful parts of an agent's acquired history. We show this recipe can be instantiated using conditional diffusion models and simple relevance functions such as curiosity- or value-based metrics. Our approach consistently improves performance and sample efficiency in both state- and pixel-based domains. We expose the mechanisms underlying these gains, showing how guidance promotes diversity in our generated transitions and reduces overfitting. We also showcase how our approach can train policies with even higher update-to-data ratios than before, opening up avenues to better scale online RL agents.[1]

## 1 INTRODUCTION

A central problem in online reinforcement learning (RL) involves extracting signal from a continuous stream of experience. Not only does the non-i.i.d. form of this data induce learning instabilities, but agents may lose out on near-term experience that is not immediately useful, but becomes important much later. The standard solution to these problems is to use a replay buffer as a form of memory (Lin, 1992; Mnih et al., 2015). By storing a dataset of transitions to enable batch-wise sampling, the algorithm can decorrelate online observations and revisit past experience later in training.

However, the idea that an agent's memory must identically reproduce past transitions, and at uniform frequency, is limiting. In general, the distribution of states an agent *visits* is different from the optimal distribution of states the agent should *train on*. Certain classes of transitions are more relevant to learning, i.e. data at critical decision boundaries or data that the agent has seen less frequently. Locating such long-tailed data is tricky, yet clearly important for efficient learning. The challenge is thus to design a scalable memory system that replays *relevant* data, and at large quantities.

In this work, we propose a simple, plug-and-play formulation of an agent's online memory that can be constructed using a generative model. We train a conditional generative model in an end-to-end manner to faithfully capture online transitions from the agent. This paradigm grants two key benefits, enabling 1) the *densification* of past experience, allowing us to create new training data that goes beyond the data distribution observed online, and 2) *guidance* via a family of "relevance functions" $\mathcal{F}$ that push these generations towards more useful parts of an agent's acquired experience.

An ideal relevance function $\mathcal{F}$ should be easy to compute, and naturally identify the most relevant experience. Intuitively, we should generate transitions at the "frontier" of the agent's experience. These are regions of transition space that our generative model can accurately capture from the agent's memory, but our policy has not yet completely mastered. We investigate a series of possible functions, concluding that *intrinsic curiosity* can successfully approximate this distribution.

---

[1]Project page available at: https://pgenreplay.github.io

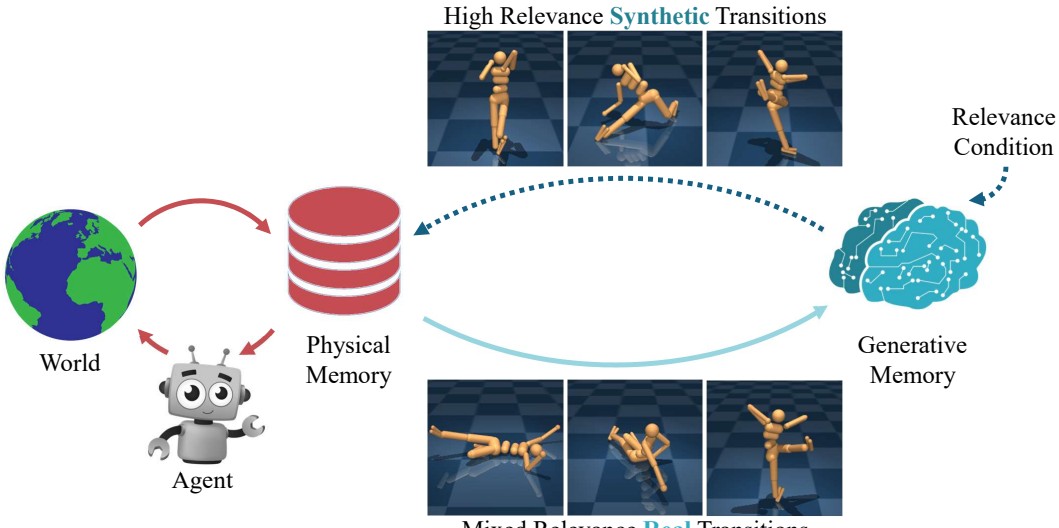

High Relevance **Synthetic** Transitions

Relevance Condition

Physical Memory

World

Agent

Generative Memory

Mixed Relevance **Real** Transitions

Figure 1: **We model an agent's online memory using a conditional diffusion model.** By conditioning on measures of data relevance, we can generate samples more useful for policy learning.

Our main contribution is this framework for a scalable, guidable generative replay, which we term "Prioritized Generative Replay" (PGR). We instantiate this framework by making use of strong diffusion model architectures. Experiments on both state-based and pixel-based RL tasks show PGR is consistently more sample-efficient than both model-free RL algorithms and generative approaches that do not use any guidance. In fact, by densifying the more relevant transitions, PGR is able to succeed in cases where unconditional generation struggles significantly. Moreover, we empirically demonstrate that PGR goes beyond simple prioritized experience replay; in particular, we show that conditioning on curiosity leads to more diverse and more learning-relevant generations. Finally, we show how PGR improves with larger policy networks, and continues learning reliably with higher synthetic-to-real data ratios, setting up a promising recipe for data-efficient scaling.

## 2 RELATED WORK

**Model-based RL.** Model-based reinforcement learning involves interacting with a predictive model of the environment, sometimes referred to as a world model (Ha & Schmidhuber, 2018), to learn a policy (Sutton, 1991). Two classes of approaches dominate: planning with the learned world model directly (Hafner et al., 2019b; Schrittwieser et al., 2020; Ye et al., 2021; Chua et al., 2018; Ebert et al., 2018), or optimizing a policy by unrolling trajectories "in the imagination" of the world model (Janner et al., 2019; 2020; Hafner et al., 2019a; Oh et al., 2017; Feinberg et al., 2018). This latter approach is most relevant to our problem setting. One distinction is we do not backprop or plan actions through a model directly, but rather wholly synthesize additional, high-relevance transitions. Closely related is PolyGRAD (Rigter et al., 2023) which generates entire on-policy trajectories. However, by generating *independent off-policy transitions* rather than attempting to approximate trajectories from the current policy, we avoid the issue of compounding errors in our model (Gu et al., 2016), and also enable easier plug-and-play with popular existing online RL methods.

**Prioritized replay.** Agents often benefit significantly more by learning from important experiences (Moore & Atkeson, 1993; Andre et al., 1997). One of the most well-known strategies which leverages this insight is prioritized experience replay (PER), which uses temporal difference (TD) error as a priority criterion to determine the relative importance between transitions (Schaul et al., 2015). Since then, a number of works have proposed many different criteria to determine transition importance, including state entropy (Ramicic & Bonarini, 2017), learnability (Sujit et al., 2023), or some combination of value/reward and TD-error (Cao et al., 2019; Gao et al., 2021). To alleviate the computational cost associated with evaluating the priority of all experiences in the replay buffer, different flavors of algorithmic improvements (Kapturowski et al., 2018; Schaul et al., 2015) or approximate parametric models of prior experience (Shin et al., 2017; Novati & Koumoutsakos, 2019) have been proposed. Our proposal to model the replay buffer as a parametric generative model is

closer to this second class of works. However, distinct from these methods which seek to relabel *existing* data, our method uses priority to generate entirely *new* data for more generalizable learning.

**RL from synthetic data.** The use of generative training data has a long history in reinforcement learning (Shin et al., 2017; Raghavan et al., 2019; Imre, 2021; Hafner et al., 2019a). But only with the recent advent of powerful diffusion models have such methods achieved parity with methods trained on real data (Janner et al., 2022; Ajay et al., 2022; Zhu et al., 2023). The use of diffusion for RL was first introduced by Diffuser (Janner et al., 2022). The authors propose an offline diffusion model for generating trajectories of states and actions, and directly generate plans which reach goal states or achieve high rewards. Decision Diffuser (Ajay et al., 2022) generates state-only trajectories, and employs an inverse kinematics model to generate corresponding actions. More recently, works like Ding et al. (2024); He et al. (2024) leverage diffusion models to augment datasets for offline RL, improving training stability. Most similar to our approach is SYNTHER (Lu et al., 2024), which augments the replay buffer online and can be seen as an unguided form of our framework. In contrast to these previous works, we posit that the strength of synthetic data models is that they can be *guided* towards novel transitions that are off-policy yet more relevant for learning.

## 3 BACKGROUND

This section offers a primer on online reinforcement learning and diffusion models. Here, details on conditional generation via guidance are especially important to our method exposition in Section 4.

**Reinforcement learning.** We model the environment as a fully-observable, infinite-horizon Markov Decision Process (MDP) (Sutton & Barto, 2018) defined by the tuple $\mathcal{M} = (\mathcal{S}, \mathcal{A}, \mathcal{P}, \mathcal{R}, p_0, \gamma)$. Here, $\mathcal{S}, \mathcal{A}$ denote the state and action spaces, respectively. $\mathcal{P}(s' \mid s, a)$ for $s, s' \in \mathcal{S}$ and $a \in \mathcal{A}$ describes the transition dynamics, which are generally not known. $\mathcal{R}(s, a)$ is a reward function, $p_0$ is the initial distribution over states $s_0$, and $\gamma$ is the discount function. In online RL, a policy $\pi : \mathcal{S} \rightarrow \mathcal{A}$ interacts with the environment $\mathcal{M}$ and observes tuples $\tau = (s, a, s', r)$ (i.e. transitions), which are stored in a replay buffer $\mathcal{D}$. The *action-value*, or $\mathcal{Q}$, function is given by:

$$Q_\pi(s, a) = \mathbb{E}_{a_t \sim \pi(\cdot|s_t), s_{t+1} \sim \mathcal{P}(\cdot|s_t, a_t)} \left[ \sum_{t=0}^{\infty} \gamma^t \mathcal{R}(s_t, a_t \mid s_0 = s, a_0 = a) \right]. \tag{1}$$

The $\mathcal{Q}$-function describes the expected *return* after taking action $a$ in state $s$, under the policy $\pi$. Then our goal is to learn the optimal policy $\pi^*$ such that $\pi^*(a \mid s) = \mathrm{argmax}_\pi Q_\pi(s, a)$.

**Conditional diffusion models.** Diffusion models are a powerful class of generative models which employ an iterative denoising procedure for generation (Sohl-Dickstein et al., 2015; Ho et al., 2020). Diffusion first involves a *forward process* $q(x^{n+1} \mid x^n)$ iteratively adding Gaussian noise to $x^n$ starting from some initial data point $x^0$. A *reverse process* $p_\theta(x^{n-1} \mid x^n)$ then transforms random Gaussian noise into a sample from the original distribution. In particular, we learn a neural network $\epsilon_\theta$ which predicts the amount of noise $\epsilon \sim \mathcal{N}(0, \mathbf{I})$ injected for some particular forward step $x_n$.

For additional controllability, diffusion models naturally enable conditioning on some signal $y$, simply by formulating the forward and reverse processes as $q(x^{n+1} \mid x^n, y)$ and $p_\theta(x^{n-1} \mid x^n, y)$, respectively. Classifier-free guidance (CFG) is a common post-training technique which further promotes sample fidelity to the condition $y$ in exchange for more complete mode coverage (Ho & Salimans, 2022). To facilitate CFG at sampling-time, during training, we optimize $\epsilon_\theta$ with the following objective:

$$\mathbb{E}_{x^0 \sim \mathcal{D}, \epsilon \sim \mathcal{N}(0, \mathbf{I}), y, n \sim \mathrm{Unif}(1, N), p \sim \mathrm{Bernoulli}(p_{\mathrm{uncond}})} \left\| \epsilon_\theta \left( x^n, n, (1 - p) \cdot y + p \cdot \varnothing \right) \right\|_2^2, \tag{2}$$

where $p_{\mathrm{uncond}}$ is the probability of dropping condition $y$ in favor of a null condition $\varnothing$. During sampling, we take a convex combination of the conditional and unconditional predictions, i.e. $\omega \cdot \epsilon_\theta(x^n, n, y) + (1 - \omega) \cdot \epsilon_\theta(x^n, n, \varnothing)$, where $\omega$ is a hyperparameter called the *guidance scale*.

## 4 PRIORITIZED GENERATIVE REPLAY

In this section, we introduce Prioritized Generative Replay (PGR). At its core, PGR involves a parametric generative replay buffer that can be guided by a variety of relevance criteria. We first provide intuition and motivation for such a framework, and concretize how it can be instantiated. Next, we compare and contrast between various instantiations of relevance functions. Most interestingly, we will empirically show that a good default choice for sample-efficient learning is *intrinsic curiosity*.

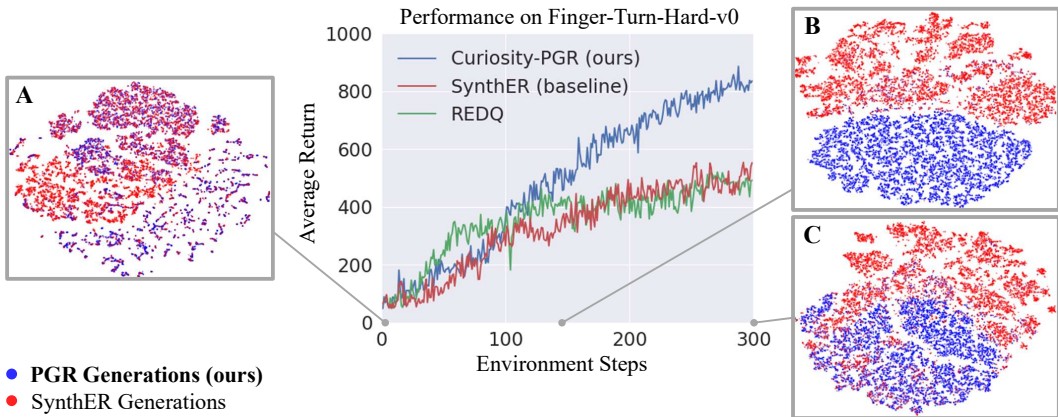

**PGR Generations (ours)**
SynthER Generations

Figure 2: **PGR improves performance by *densifying* subspaces of data where transitions more relevant for learning reside.** We project 10K generations for both our PGR and the unconditional baseline SYNTHER to the same tSNE plot. **A:** At epoch 1, the distribution of data generated by PGR and SYNTHER are similar. **B:** At the inflection point of performance near epoch 130, PGR generates a distinct sub-portion of the data space from SYNTHER (i.e. red and blue dots are largely separate.) **C:** At the end of learning, PGR still densely covers a distinct subspace of the SYNTHER transitions.

## 4.1 MOTIVATION

Consider an agent with policy $\pi$ interacting online with some environment $\mathcal{M}$. The agent observes and stores transitions $\tau = (s, a, s', r)$ in some finite replay buffer $\mathcal{D}$. Two main issues arise:

1. $\mathcal{D}$ must be sufficiently large and diverse to prevent overfitting (hard to guarantee online)

2. Transitions more relevant for updating $\pi$ might be rare (and thus heavily undersampled)

To address the first problem, we can *densify* the replay distribution $p_{\mathcal{D}}(\tau)$ by learning a generative world model using the buffer. This effectively trades in our finitely-sized, non-parametric replay buffer for an infinitely-sized, parametric one. By leveraging the generalization capabilities of, e.g., diffusion models, we can interpolate the replay distribution to more impoverished regions of data.

However, uniformly sampling from this parametric buffer, in expectation, implies simply replaying past transitions at the same frequency at which they were observed. In more challenging environments, there may only a small fraction of data which is most relevant to updating the current policy $\pi$. Thus, to address the second problem and enable sample-efficient learning, we need some mechanism to not only *densify* $p_{\mathcal{D}}(\tau)$, but actually *guide* it towards the more immediately useful set of transitions.

Our key insight is to frame this problem via the lens of conditional generation. Specifically, we seek to model $p_{\mathcal{D}}(\tau \mid c)$, for some condition $c$. By choosing the right $c$, we "marginalize out" parts of the unconditional distribution $p_{\mathcal{D}}(\tau)$ which are less relevant under $c$ (see Fig. 2.) More concretely, we propose to learn a *relevance function* $\mathcal{F}(\tau) = c$ jointly with the policy $\pi$. Intuitively, this relevance function measures the "priority" $c$ of $\tau$. Overall, we can achieve both more *complete* as well as more *relevant* coverage of $p_{\mathcal{D}}(\tau)$. Clearly, the choice of $\mathcal{F}$ in our framework is critical. We now explore a number of instantiations for $\mathcal{F}$, and perform an analysis on their strengths and weaknesses.

## 4.2 RELEVANCE FUNCTIONS

We begin by outlining two desiderata for our relevance functions. First, given our online setting, these functions should incur minimal computation cost. This removes conditioning on strong priors which demand extensive pretraining (e.g. prior works in offline RL (Du & Narasimhan, 2019; Schwarzer et al., 2021)). Second, we want a guidance signal that will not overfit easily, thereby collapsing generation to stale or low-quality transitions even as the agent's experience evolves over time.

**Return.** Our first proposal is one of the most natural: we consider a relevance function based on episodic return. Concretely, we use the value estimate of the learned $Q$-function and current policy $\pi$:

$$\mathcal{F}(s, a, s', r) = Q(s, \pi(s)). \tag{3}$$

---

**Algorithm 1** Overview of our outer loop + inner loop framework.

---

**Input:** synthetic data ratio $r \in [0, 1]$, conditional guidance scale $\omega$
**Initialize:** $\mathcal{D}_{\text{real}} = \varnothing$ real replay buffer, $\pi$ agent, $\mathcal{D}_{\text{syn}} = \varnothing$ synthetic replay buffer, $G$ generative model, "relevance function" $\mathcal{F}$
1: **while** forever **do** ▷ perpetual outer loop
2:      Collect transitions $\tau_{\text{real}}$ with $\pi$ in the environment and add to $\mathcal{D}_{\text{real}}$
3:      Update $\mathcal{F}$ using $\mathcal{D}_{\text{real}}$ and Eqs. (3) to (5)
4:      **for** 1, ..., T **do** ▷ periodic inner loop
5:          Sample $\tau_{\text{real}}$ from $\mathcal{D}_{\text{real}}$ and optimize $G(\tau \mid \mathcal{F}(\tau))$ using Eq. (2)
6:          Conditionally generate $\tau_{\text{syn}}$ from $G$ with guidance scale $\omega$ and add to $\mathcal{D}_{\text{syn}}$
7:          Train $\pi$ on samples from $\mathcal{D}_{\text{real}} \cup \mathcal{D}_{\text{syn}}$ mixed with ratio $r$
8:      **end for**
9: **end while**

---

Return-as-relevance sensibly pushes generations to be more on-policy, since $\pi$ by construction seeks out high $Q$-estimate states. Also, many online RL algorithms already learn a $Q$-function, and so we readily satisfy the first condition (Mnih et al., 2015). However, the second condition is not adequately addressed by this choice of $\mathcal{F}$. In particular, the diversity of high-return transitions might in practice be quite low, making overfitting to the conditional generations more likely.

**Temporal difference (TD) error.** Another possible choice for our relevance function is TD-error, first proposed for replay prioritization by Schaul et al. (2015). Our relevance function in this case is given by the difference between the current Q-value estimate and the bootstrapped next-step estimate:

$$\mathcal{F}(s, a, s', r) = r + \gamma Q_{\text{target}}\left(s', \underset{a'}{\arg\max} Q(s', a')\right) - Q(s, a), \tag{4}$$

One immediate shortcoming is that in practice we have no guarantees on the $Q$-estimates for rarer, out-of-distribution transitions. In fact, overly greedy prioritization of high TD-error transitions can lead to low-quality, myopic $Q$-estimates (Schaul et al., 2015).

**Curiosity.** A glaring problem with both return-based and TD error-based relevance functions is their reliance on high-quality $Q$-functions. Estimation errors can thus lead to $\mathcal{F}$ providing a poor conditioning signal. Moreover, online RL agents tend to overfit $Q$-functions to early experience (Nikishin et al., 2022), which will in turn lead to a rapidly overfitted $\mathcal{F}$ under these two choices.

Naturally then, we might consider reducing overfitting via some relevance function which promotes generation *diversity*. As prior work has shown, an effective way to decrease overfitting to early, noisy signal in online RL is to leverage diverse experience (Zhang et al., 2018). To achieve this diversity, we model $\mathcal{F}$ after exploration objectives which promote engaging with "higher-novelty" transitions that are more rarely seen (Strehl & Littman, 2008). Moreover, by learning a separate function entirely, we decorrelate our relevance function from the $Q$-function, making overfitting less likely.

We thus turn to prior work on intrinsic motivation (Schmidhuber, 1991; Oudeyer & Kaplan, 2007) to operationalize these insights. Concretely, we take inspiration from the intrinsic curiosity module (Pathak et al., 2017) to parameterize $\mathcal{F}$. Given a feature encoder $h$, we learn a forward dynamics model $g$ which models the environment transition function $\mathcal{P}(s' \mid s, a)$, in the latent space of $h$. Then $\mathcal{F}$ is given by the *error* of this forward dynamics model:

$$\mathcal{F}(s, a, s', r) = \frac{1}{2}\|g(h(s), a) - h(s')\|^2. \tag{5}$$

### 4.3 PGR FRAMEWORK SUMMARY

Finally, we provide a concrete overview of our framework in Algorithm 1. In the outer loop, the agent interacts with the environment, receiving a stream of real data and building a replay buffer $\mathcal{D}_{\text{real}}$, as in regular online RL. In the event that we are using a curiosity-based relevance function, we also perform an appropriate gradient update for $\mathcal{F}$ using samples from $\mathcal{D}_{\text{real}}$, via the loss function given by Eq. (5). Then periodically in the inner loop, we learn a conditional generative model $G$ of $\mathcal{D}_{\text{real}}$ and generatively densify these transitions to obtain our synthetic replay buffer $\mathcal{D}_{\text{syn}}$. Concretely, we take $G$ to be a conditional diffusion model. To leverage the conditioning signal given by $\mathcal{F}$, we use

| | DMC-100k (Online) | | | | Pixel-DMC-100k (Online) | |
| --- | --- | --- | --- | --- | --- | --- |
| Environment | Quadruped-Walk | Cheetah-Run | Reacher-Hard | Finger-Turn-Hard* | Walker-Walk | Cheetah-Run |
| MBPO | 505.91 ± 252.55 | 450.47 ± 132.09 | 777.24 ± 98.59 | 631.19 ± 98.77 | - | - |
| DREAMER-V3 | 389.63 ± 168.47 | 362.01 ± 30.69 | 807.58 ± 156.38 | 745.27 ± 90.30 | 353.40 ± 114.12 | 298.13 ± 86.37 |
| SAC | 178.31 ± 36.85 | 346.61 ± 61.94 | 654.23 ± 211.84 | 591.11 ± 41.44 | - | - |
| REDQ | 496.75 ± 151.00 | 606.86 ± 99.77 | 733.54 ± 79.66 | 520.53 ± 114.88 | - | - |
| REDQ + CURIOSITY | 687.14 ± 93.12 | 682.64 ± 52.89 | 725.70 ± 87.78 | 777.66 ± 116.96 | - | |
| DRQ-V2 | - | - | - | - | 514.11 ± 81.42 | 489.30 ± 69.26 |
| SYNTHER | 727.01 ± 86.66 | 729.35 ± 49.59 | 838.60 ± 131.15 | 554.01 ± 220.77 | 468.53 ± 28.65 | 465.09 ± 28.27 |
| PGR (Reward) | 510.39 ± 121.11 | 660.87 ± 87.54 | 715.43 ± 97.56 | 540.85 ± 73.29 | - | - |
| PGR (Return) | 737.62 ± 20.13 | 779.42 ± 30.00 | 893.65 ± 55.71 | 805.42 ± 92.07 | - | - |
| PGR (TD Error) | 802.18 ± 116.52 | 704.17 ± 96.49 | **917.61 ± 37.32** | 839.26 ± 49.90 | - | - |
| PGR (Curiosity) | **927.98 ± 25.18** | **817.36 ± 35.93** | 915.21 ± 48.24 | **885.98 ± 67.29** | 570.99 ± 41.44 | 529.70 ± 27.76 |

Table 1: **Average returns on state and pixel-based DMC after 100K environment steps (5 seeds, 1 std. dev. err.).** * is a harder environment with sparser rewards, and so we present results over 300K timesteps.

| | Walker2d-v2 | HalfCheetah-v2 | Hopper-v2 |
| --- | --- | --- | --- |
| MBPO | 3781.34 ± 912.44 | 8612.49 ± 407.53 | 3007.83 ± 511.57 |
| DREAMER-V3 | 4104.67 ± 349.74 | 7126.84 ± 539.22 | 3083.41 ± 138.90 |
| SAC | 879.98 ± 217.52 | 5065.61 ± 467.73 | 2033.39 ± 793.96 |
| REDQ | 3819.17 ± 906.34 | 6330.85 ± 433.47 | 3275.66 ± 171.90 |
| SYNTHER | 4829.32 ± 191.16 | 8165.35 ± 1534.24 | 3395.21 ± 117.50 |
| PGR (Curiosity) | **5682.33 ± 370.04** | **9234.61 ± 658.77** | **4101.79 ± 244.05** |

Table 2: **Results on state-based OpenAI gym tasks.** We report average return after 100K environment steps. Results are over 3 seeds, with 1 std. dev. err.

| | REDQ | SYNTHER | PGR |
| --- | --- | --- | --- |
| Model Size (x$10^6$ params.) | 9.86 | 7.12 | 7.39 (+3.7%) |
| Generation VRAM (GB) | - | 4.31 | 6.67 (+54.7%) |
| Train Time, hours (Diffusion) | - | 1.57 | 1.61 (+2.5%) |
| Generation Time, hours | - | 0.62 | 0.63 (+1.6%) |
| Train Time, hours (RL) | 5.69 | 1.98 | 2.11 (+6.5%) |
| Train Time, hours (Total) | 5.69 | 4.17 | 4.35 (+4.3%) |

Table 3: **Model runtime and latency.** PGR incurs <5% additional training time compared to SynthER. Generation also fits easily on modern GPUs (<12GB).

CFG and a "prompting" strategy inspired by Peebles et al. (2022). We choose some ratio $k$ of the transitions in the real replay buffer $\mathcal{D}_{\text{real}}$ with the highest values for $\mathcal{F}(s, a, s', r)$, and sample their conditioning values randomly to pass to $G$. Implementation-wise, we keep both $\mathcal{D}_{\text{real}}$ and $\mathcal{D}_{\text{syn}}$ at 1M transitions, and randomly sample synthetic and real data mixed according to some ratio $r$ to train our policy $\pi$. Here, any off-policy RL algorithm can be used to learn $\pi$. For fair comparison to prior art, we instantiate our framework with both SAC (Haarnoja et al., 2018) and REDQ (Chen et al., 2021).

## 5 EXPERIMENTS

In this section, we answer three main questions. (1) First, to what extent does our PGR improve performance and sample efficiency in online RL settings? (2) Second, what are the underlying mechanisms by which PGR brings about performance gains? (3) Third, what kind of scaling behavior does PGR exhibit, if indeed conditionally-generated synthetic data under PGR is better?

**Environment and tasks.** Our results span a range of state-based and pixel-based tasks in the DeepMind Control Suite (DMC) (Tunyasuvunakool et al., 2020) and OpenAI Gym (Brockman, 2016) environments. In particular, our benchmark follows exactly the online RL evaluation suite of prior work in generative RL by Lu et al. (2024), facilitating direct comparison. In all tasks, we allow 100K environment interactions, a standard choice in online RL (Li et al., 2023; Kostrikov et al., 2020).

**Model details, training and evaluation.** For state-based tasks, in addition to the SAC and REDQ model-free baselines, we also include two strong model-based online RL baselines in MBPO (Janner et al., 2019) and DREAMER-V3 (Hafner et al., 2023). We also compare against SYNTHER (Lu et al., 2024), a recent work which learns an unconditional generative replay model, allowing SAC to be trained with an update-to-data (UTD) ratio of 20. To facilitate conditional sampling, during training we randomly discard the scalar given by our relevance function $\mathcal{F}$ with probability 0.25.

For pixel-based tasks, our policy is based on DRQ-V2 (Yarats et al., 2021) as in Lu et al. (2022). To maintain the same approach and architecture for our generative model, we follow Lu et al. (2024); Esser et al. (2021) and generate data in the latent space of the policy's CNN visual encoder. That is, given a visual encoder $f_\theta$, and a transition $(s, a, s', r)$ for pixel observations $s, s' \in \mathbb{R}^{3 \times h \times w}$ of height $h$ and width $w$, we learn to (conditionally) generate transitions $(f_\theta(s), a, f_\theta(s'), r)$.

In the interest of fair comparison, our diffusion and policy architectures mirror SYNTHER exactly. Thus, training FLOPs, parameter count and generation time as presented in Table 3 are all directly comparable to SYNTHER. Our only additional parameters lie in a lightweight curiosity head, which is updated for only 5% of all policy gradient steps. PGR thus incurs minimal additional overhead.

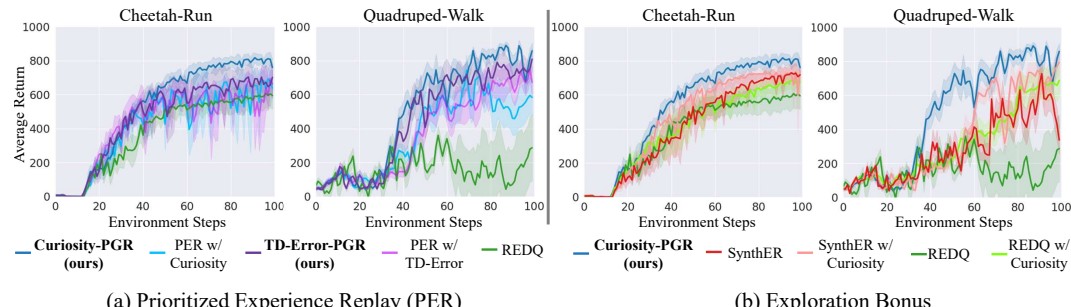

(a) Prioritized Experience Replay (PER)

(b) Exploration Bonus

Figure 3: **Comparison to baselines that use (a) prioritized experience replay (PER) and (b) exploration reward bonuses.** REDQ using PER, with priority determined by curiosity Eq. (5) or TD-error Eq. (4), still underperform their PGR counterparts. PGR also remains superior after directly adding an exploration bonus in the form of curiosity to either SYNTHER or REDQ.

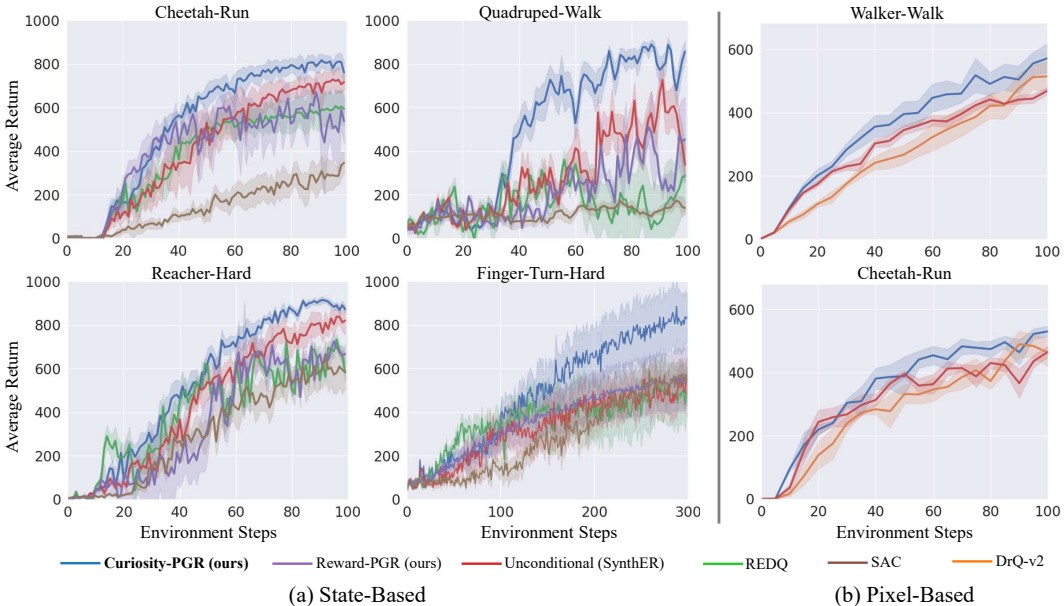

(a) State-Based

(b) Pixel-Based

Figure 4: **Sample efficiency on DMC (a) state-based and (b) pixel-based tasks.** We show mean and standard deviation over five seeds. Curiosity-PGR consistently demonstrates the best sample efficiency. SYNTHER, which uses unconditional generation to augment replay, underperforms model-free algorithms like SAC and REDQ on harder sparse-reward tasks, like finger-turn-hard.

## 5.1  RESULTS

As shown in Table 1 and Table 2, all variants of PGR successfully solve the tasks, with curiosity guidance consistently outperforming both model-free (SAC, REDQ and DRQ-V2), and model-based (MBPO and DREAMER-V3) algorithms, as well as unconditional generation (SYNTHER). We also investigate alternative measures of curiosity such as random network distillation (Burda et al., 2018) and episodic curiosity (Savinov et al., 2018) in stochastic environments, with results in Appendix A.

**Comparison to prioritized experience replay (PER).** We also compare PGR to PER baselines that use different measures of priority. In particular, we train REDQ with a prioritized replay buffer, using the classic TD-error (Schaul et al., 2015), or a priority function based on Eq. (5). The former is compared to our PGR approach using Eq. (4) as relevance, and the latter to using Eq. (5). As shown in Fig. 3a, PGR demonstrates superior performance in both cases. This emphasizes the importance of *densifying* the replay distribution with generations, and not simply reweighting past experience.

**Comparison to exploration bonuses.** We examine how baselines improve when given a bonus in the form of an intrinsic curiosity reward (*c.f.* Eq. (5)). In Fig. 3b, we see that curiosity-PGR continues to outperform SYNTHER or REDQ when either is augmented this way. This suggests PGR goes

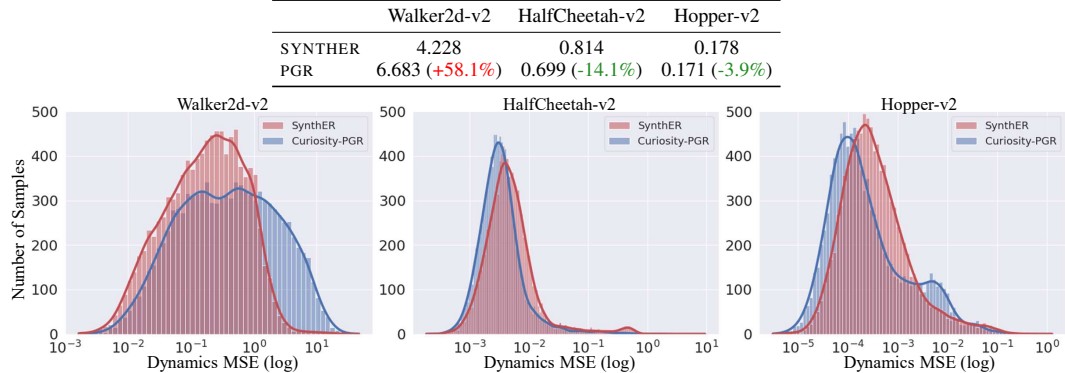

|          | Walker2d-v2     | HalfCheetah-v2  | Hopper-v2      |
|----------|-----------------|-----------------|----------------|
| SYNTHER  | 4.228           | 0.814           | 0.178          |
| PGR      | 6.683 (+58.1%)  | 0.699 (-14.1%)  | 0.171 (-3.9%)  |

Figure 5: **PGR does not outperform baselines due to improved generation quality.** We compute mean-squared error (MSE) of dynamics over 10K generated transitions for SYNTHER and curiosity-PGR across 3 OpenAI gym environments. **Top:** Average MSE. **Bottom:** Histograms of MSE values.

beyond just improving exploration. We provide additional evidence for this claim in Appendix B. We propose our gains are a result of generating higher novelty transitions from the replay buffer, with higher diversity. This additional diversity thereby reduces overfitting of the $Q$-function to synthetic transitions. We provide further evidence for this idea in Section 5.2.

## 5.2 SAMPLE EFFICIENCY

In this section, we show PGR achieves its strong performance in a sample-efficient manner, and reveal the mechanisms underlying this sample efficiency. As seen in Fig. 4a, for tasks with state-based inputs, SYNTHER often attains its best performance around 100K environment steps. In contrast, curiosity-PGR is able to match this performance in both the cheetah-run and quadruped-walk tasks after only ~50K steps. Especially noteworthy is the finger-turn-hard task, where SYNTHER actually underperforms compared to the vanilla model-free REDQ baseline, while our curiosity-PGR continues outperforming. Guidance is particularly important in sparse reward tasks, where decision-critical data is also naturally sparser. These observations hold for pixel-based tasks. As we see in Fig. 4b, while SYNTHER is eventually overtaken by DRQ-V2 in both environments, our curiosity-PGR continues to consistently improve over DRQ-V2.

**Is the solution to simply replace unconditional with conditional generation?** Conditional generation is well-known to improve sample quality in the generative modeling literature (Ho & Salimans, 2022; Chen et al., 2023). Thus, one sensible assumption is that PGR exhibits stronger performance simply because our generated samples are better in quality. We show this is not the case.

Specifically, we borrow the methodology of Lu et al. (2024) and measure faithfulness of generated transitions to environment dynamics. Given a generated transition $(s, a, s', r)$, we roll out the action $a$ given the current state $s$ in the environment simulator to obtain the ground truth next state and reward. We then measure the mean-squared error (MSE) on these two values, comparing against generated next state $s'$ and reward $r$, respectively. This analysis is performed at epoch 50 (halfway through online policy learning), over 10K generated transitions, and across 3 different environments.

As seen in Fig. 5, SYNTHER and PGR are highly similar in terms of generation quality. Thus, our motivations for using conditional generation have nothing to do with traditional arguments in generative modeling surrounding enhanced generation quality. Generations under PGR are better because what matters is not simply quality, but generating the *right classes/kinds of transitions*. More generally, our core contribution is to elegantly cast widely-used prioritized replay in online RL through this lens of conditional generation.

Moreover, naïve choices for conditioning fail entirely. For example, training on a larger quantity of higher reward transitions should intuitively yield a high-reward policy. However, as we show in Fig. 4 and Table 1, naïvely conditioning on high reward (Reward-PGR) actually results in worse performance than any of the other PGR variants we propose, and in fact does worse than unconditional generation (SYNTHER). This is because greedily densifying high reward transitions does not actually provide the policy with any data on how to navigate to those high reward environment regions. Thus, for obtaining strong performance, it is important to be thoughtful in our choice of $\mathcal{F}$.

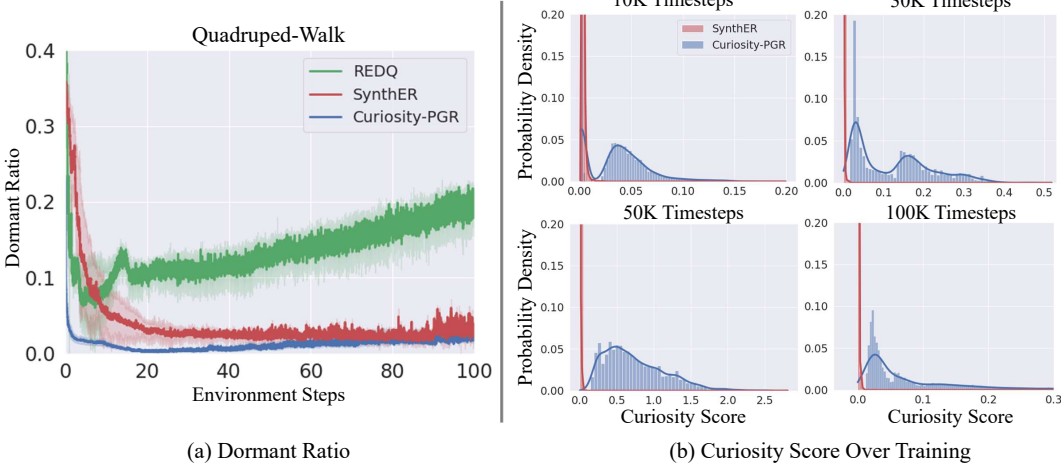

(a) Dormant Ratio      (b) Curiosity Score Over Training

Figure 6: **Curiosity-PGR reduces overfitting and improves diversity of replay data.** (a) Dormant ratio (Xu et al., 2023) (DR) of policy networks for different approaches. DR is consistently lower for PGR, indicating a minimally overfitting policy. (b) Curiosity $\mathcal{F}$-values throughout training for the unconditional baseline and PGR. We show that the distribution of states accessed by the curiosity-PGR policy is significantly shifted towards higher novelty environment transitions.

**What is the mechanism underlying our performance gains?** We now validate our argument in Section 4.2 for conditioning on relevance functions which mitigate overfitting. We quantify the "dormant ratio" (DR) (Sokar et al., 2023) over the course of learning on the quadruped-walk task. DR is the fraction of inactive neurons in the policy network (i.e. activations below some threshold). Prior work has shown this metric effectively quantifies overfitting in value-based RL, where higher DR correlates with policies that execute unmeaningful actions (Sokar et al., 2023; Xu et al., 2023).

As we see in Fig. 6a, the REDQ baseline exhibits high and increasing DR over training, indicating aggressive overfitting. This reflects the underperformance of REDQ on quadruped-walk. Crucially, our curiosity-PGR displays a low and stable DR, which increases only marginally (after task performance has saturated, as seen in Fig. 4.) Moreover, our DR remains consistently below that of the unconditional SYNTHER baseline. This further suggests our PGR generates higher-relevance transitions with more diversity, and better addresses the issue of overfitting $Q$-functions to the synthetic data.

**How does curiosity lead to PGR's reduction in overfitting?** To show that conditioning on curiosity contributes to mitigating overfitting, we characterize the signal we obtain from $\mathcal{F}$ over time. Specifically, we examine the curiosity-PGR variant on the quadruped-walk task, measuring the distribution of $\mathcal{F}(s, a, s', r)$ using Eq. (5) over 10K *real* transitions. We perform this evaluation every 10K timesteps, which is the frequency at which we retrain our generative model. For comparison, we also evaluate this trained $\mathcal{F}$ on 10K real transitions encountered by the unconditional SYNTHER baseline.

As we see in Fig. 6b, as early as 10K timesteps, immediately after the policy has been trained on the first batch of synthetic data, the distribution of curiosity values is more left-skewed for the conditional model. This distribution becomes increasingly long-tailed over training, suggesting a growing diversity in the observed states as training progresses. This is contrasted by the increased biasedness of the unconditional distribution towards low-curiosity transitions. Thus, the agent trained on synthetic data from PGR is learning to move towards environment states that are more "novel" over time, improving diversity in both the real replay buffer $\mathcal{D}_{\text{real}}$ as well as synthetic replay buffer $\mathcal{D}_{\text{syn}}$. As further confirmation, we see that 100K timesteps, the environment is relatively well-explored, and curiosity values from $\mathcal{F}$ diminish, which correlates with the task reward saturation that we observe. We conclude that this is ultimately the mechanism underlying the improved sample efficiency of PGR.

## 5.3 SCALING PROPERTIES

Finally, we perform a series of experiments examining the scaling behavior of our PGR, in comparison to the unconditional SYNTHER baseline. For baselines, we reduce the capacity of the diffusion model in both PGR and SYNTHER such that the resultant policies attain the same performance as the model-free REDQ baseline on the quadruped-walk task in DMC-100K (*c.f.* dashed lines in Fig. 7). We

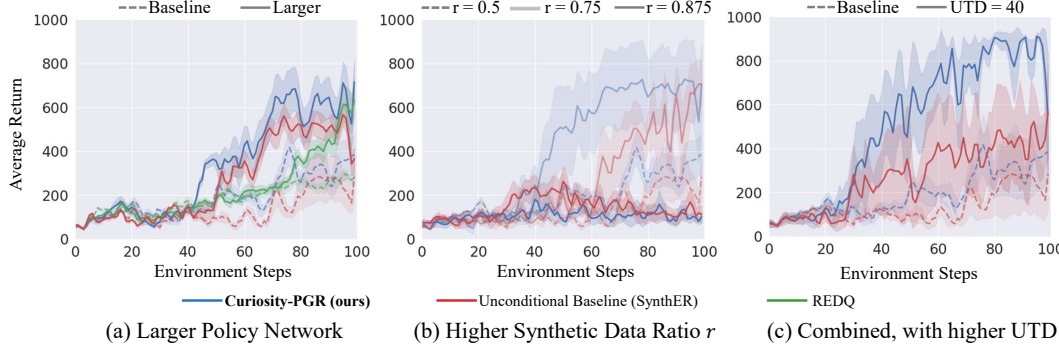

(a) Larger Policy Network     (b) Higher Synthetic Data Ratio $r$     (c) Combined, with higher UTD

Figure 7: **Scaling behavior on DMC-100K quadruped-walk.** PGR can effectively combine (a) larger networks with (b) higher ratios of synthetic data, (c) allowing us to train with a much higher UTD of 40. In comparison, the unconditional SYNTHER does not scale as well, and combining (a) and (b) actually underperforms using either independently. Runs shown are averaged over 3 seeds.

then introduce a sequence of experiments — first increasing the size of the policy network, holding synthetic data constant, then increasing both the amount of synthetic data as well as how aggressively we rely on this data for training. Results show that PGR leverages conditionally generated synthetic data in a more scalable fashion than SYNTHER is able to use its unconditionally generated data.

**Network size.** We employ the experimental protocol of SYNTHER (Lu et al., 2024): for both networks, we increase the number of hidden layers from 2 to 3, and their widths from 256 to 512. This results in ~6x more parameters, so we also increase batch size from 256 to 1024 to maintain per-parameter throughput. We see in Fig. 7a that this improves both PGR and SYNTHER, commensurately with REDQ trained with real data. This affirms that the synthetic data is a reasonable stand-in for real data.

**Amount of generative data.** Next, we analyze the behavior of PGR and SYNTHER when varying the *fraction* of generative data used per batch. This better answers the full extent to which synthetic data can replace real data. Recall our baseline models use a batch size of 256 and a synthetic data ratio $r$ of 0.5 (i.e. every batch has 128 real and 128 generated transitions). We now double the batch size to 512 and then to 1024, each time scaling $r$ to 0.75 and 0.875, respectively. This keeps the number of real transitions per-batch fixed at 128, increasing only the number of generated transitions used.

We hypothesize that our generated transitions conditioned on curiosity are more effective for learning than SYNTHER's unconditional synthetic transitions. Indeed, we observe in Fig. 7b that at $r = 0.75$, PGR continues to enjoy sample-efficient learning, whereas SYNTHER fails to improve as significantly. Surprisingly, we find that both PGR and SYNTHER fail catastrophically when the synthetic data ratio is pushed to 0.875. While useful as a mechanism to augment past online experience, further work is needed to completely supplant real environment interactions with synthetic ones.

**Merging insights.** Finally, we combine the above two insights, and show that this allows us to push the UTD ratio of PGR to new heights. For both SYNTHER and PGR, we again use larger MLPs to parameterize the policies, and a synthetic data ratio of 0.75 and batch size of 512. Finally, we double the UTD from 20 to 40, and the size of the synthetic data buffer $\mathcal{D}_{\text{syn}}$, from 1M to 2M transitions. The idea here is to preserve average diversity of the sampled synthetic transitions. We show in Fig. 7c PGR clearly leverages both components — larger networks and more generative data — in a complementary fashion to scale more effectively without additional real interactions. In contrast, SYNTHER actually degrades in performance compared with using either component independently.

## 6 CONCLUSION

In this work, we propose Prioritized Generative Replay (PGR): a parametric, prioritized formulation of an online agent's memory based on an end-to-end learned conditional generative model. PGR conditions on a relevance function $\mathcal{F}$ to guide generation towards more learning-informative transitions, improving sample efficiency in both state- and pixel-based tasks. We show that it is not conditional generation itself, but rather conditioning on the correct $\mathcal{F}$ that is critical. We identify curiosity as a good default choice for $\mathcal{F}$, and decisively answer why: curiosity-PGR improves the diversity of generative replay, and reduces overfitting to synthetic data. PGR also showcases a promising new formula for scalable training with synthetic data, opening up new directions in generative RL.

ACKNOWLEDGMENTS

The authors would like to thank Qiyang Li for discussion on experimental design for Section 5.2. The authors would also like to thank Qianqian Wang, Yifei Zhou, Yossi Gandelsman and Alex Pan for helpful edits of prior drafts. Amy Lu and Chung Min Kim also provided comments on Fig. 1. RW is supported in part by the NSERC PGS-D Fellowship (no. 587282), the Toyota Research Institute and ONR MURI. KF is supported in part by an National Science Foundation Fellowship for KF, under grant No. DGE 2146752. Any opinions, findings, and conclusions or recommendations expressed in this material are those of the author(s) and do not necessarily reflect the views of the NSF. PA holds concurrent appointments as a Professor at UC Berkeley and as an Amazon Scholar. This paper describes work performed at UC Berkeley and is not associated with Amazon.

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

## A  ADDITIONAL RELEVANCE FUNCTIONS AND THE NOISY-TV PROBLEM

We show that PGR can be instantiated using a variety of other relevance functions $\mathcal{F}$, solidifying the claim that PGR is a framework level contribution where we can flexibly condition on a wide range of $\mathcal{F}$. In particular, we examine other functions which describe curiosity-based metrics that are more robust than what is offered by the prediction error-based intrinsic curiosity of Pathak et al. (2017).

### A.1  CONDITIONING ON OTHER RELEVANCE FUNCTIONS

We begin by expanding the pixel-based DMC-100K benchmark in Table 1 to include two additional results: conditioning on error obtained via random network distillation (Burda et al., 2018), and on pseudo-counts obtained via a context tree switching switching density model (Bellemare et al., 2016).

**Random Network Distillation (RND).** RND involves randomly initializing two identical neural networks to embed observations: a fixed *target* $f$ network and a trainable *predictor* network $\hat{f}_\theta$ (Burda et al., 2018). The parameters of $\hat{f}$ are updated via minimizing the MSE with respect to the target network outputs. This optimization error then becomes exactly the relevance function we adopt.

Specifically, we parameterize the neural networks as three-layer CNNs, with bottleneck latent dimension 64 and feature output dimension 512. The CNNs are followed by a two-layer MLP projection, also of dimension 512. The relevance function can then be described as:

$$\mathcal{F}(s, a, s', r) = \frac{1}{2}\|\hat{f}_\theta(s') - f(s')\|^2.$$ (6)

Other training and model details, as well as the environment details specific to pixel-based tasks, follow our description in Section 5.

**Context-Tree Switching (CTS) Density Model.** Pseudo-counts estimate the novelty of a given state via the frequency of visits to this state (Bellemare et al., 2016). Following Theorem 1 of Bellemare et al. (2016), and the prior work of Strehl & Littman (2008), we set our relevance function as:

$$\mathcal{F}(s, a, s', r) = \left(\hat{N}(s, a) + 0.01\right)^{-\frac{1}{2}}.$$ (7)

where the learned pseudo-count $\hat{N}(s, a)$ has a closed form per Equation 2 in Bellemare et al. (2016). Specifically, $\hat{N}$ depends only on a learned density model $\rho$ over observations in state-action space. Following prior work, we implement $\rho(s, a)$ as a context tree switching (CTS) density model (Bellemare et al., 2016). We resize the visual observations to $42 \times 42$ pixels, and use 8 context bins. As with our RND variant, remaining training, task and model details follow our description in Section 5.

**Conclusion:** As observed in Table 4, PGR can effectively condition on other relevance functions to bring about similar benefits on our pixel-based online DMC-100K benchmark. However, one reasonable remaining question is how PGR might fare in more complex environments. Of particular interest are environments rife with stochastic transitions. which first motivated works like Burda et al. (2018) and Bellemare et al. (2016).

|  | Walker-Walk | Cheetah-Run |
|---|---|---|
| DRQ-V2 | $514.11 \pm 81.42$ | $489.30 \pm 69.26$ |
| SYNTHER | $468.53 \pm 28.65$ | $465.09 \pm 28.27$ |
| PGR (RND) | $\mathbf{602.10 \pm 43.44}$ | $512.58 \pm 23.81$ |
| PGR (CTS) | $540.78 \pm 88.27$ | $508.17 \pm 74.03$ |
| PGR (CURIOSITY) | $570.99 \pm 41.44$ | $\mathbf{529.70 \pm 27.76}$ |

Table 4: **Average returns on pixel-based DMC after 100K environment steps (5 seeds, 1 std. dev. err.).** We now include results using relevance functions based on RND and pseudo-counts.

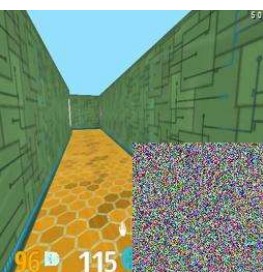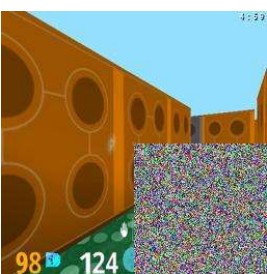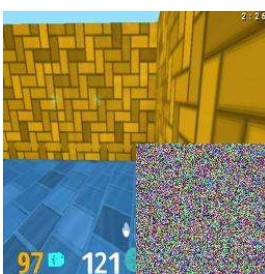

Figure 8: **Random RGB observations from the randomized *DMLab* environment.** Stochastic noise is added to the lower right quadrant, i.i.d across pixels and timesteps.

## A.2 ADDRESSING THE NOISY-TV PROBLEM

We now turn to an environment where a literal noisy TV is present. In particular, we examine the randomized *DMLab* environments presented in Section S6 of Savinov et al. (2018). These environments consist of procedurally generated minigrid 3D mazes, with 9 discrete actions available at a repeat frequency of 4. Observations are first-person $84 \times 84$ RGB images of the agent's current view within the maze. Reward is sparse (+10 for reaching the goal, 0 at all other times), and the agent is required to reach the goal as many times as possible within 1800 environment steps. We consider two variants of this task: "Sparse" is the default environment just described, and "Very Sparse" is a harder version where same-room initializations that falsely give immediate reward are removed.

Importantly, stochasticity in observations is injected by replacing the lower right $42 \times 42$ pixels of the agent's view with noise sampled uniformly from $[0, 255]$, independently for each pixel. A visualization of this input is available in Fig. 8.

We reproduce many of the same baselines as in Savinov et al. (2018), and also introduce our own. Namely, a base agent is trained using a vanilla PPO implementation (Schulman et al., 2017). We also introduce additional variants which augment the extrinsic reward with intrinsic curiosity (either ICM or RND), as well episodic curiosity (ECO) introduced by Savinov et al. (2018). Intrinsic bonus scales, learning rates and other hyperparameters follow Table S3 of Savinov et al. (2018).

We also adapt PGR to the *DMLab* environment. For fair comparison, we swap out the model-free REDQ backbone with PPO, but preserve the residual MLP denoising diffusion model described in Section 5. We use relevance functions based on ICM (*c.f.* Eq. (5)) and RND (*c.f.* Eq. (6)).

**Combining PGR with ECO.** Crucially, we also show that PGR can condition on a relevance function derived from ECO, *with no hyperparameter tuning*. ECO operates by characterizing novelty via *reachability* — that is, a particular state is novel if and only if it cannot be reached within some $k$ actions of observations in a memory buffer $\mathbf{M}$. To estimate reachability, ECO embeds observations using an embedding network $E$ and compares them to other embeddings within $\mathbf{M}$ using a comparator $\mathcal{C}$, trained via a logistic regression loss. The output of $\mathcal{C}$ is close to 1 if the probability of two observations being "close" to each other is high, and 0 otherwise. Given these elements, we define our relevance function via Equation 3 in Savinov et al. (2018):

$$\mathcal{F}(s, a, s', r) = \alpha \left( \beta - F(\mathcal{C}(E(s), E(s_i))) \right) \quad \forall s_i \in \mathbf{M} \tag{8}$$

|  | Sparse | Very Sparse |
|---|---|---|
| PPO | $7.3 \pm 2.7$ | $4.1 \pm 2.4$ |
| PPO + CURIOSITY | $5.6 \pm 1.8$ | $2.8 \pm 1.0$ |
| PPO + RND | $8.2 \pm 1.9$ | $4.0 \pm 1.1$ |
| PPO + ECO | $16.3 \pm 3.6$ | $12.9 \pm 1.9$ |
| PGR (CURIOSITY) | $9.3 \pm 2.0$ | $5.7 \pm 2.2$ |
| PGR (RND) | $11.2 \pm 1.0$ | $8.0 \pm 1.8$ |
| PGR (ECO) | $\mathbf{21.9 \pm 2.6}$ | $\mathbf{18.7 \pm 2.1}$ |

Table 5: **Average returns on randomized *DMLab* after 10M env. steps (10 seeds, 1 std. dev. err.).**

We directly use the recommended hyperparameters in Savinov et al. (2018), setting $\alpha = 0.03$, $\beta = 0.5$, $|\mathbf{M}| = 200$, and $F$ = percentile-90. The embedder network $E$ is a ResNet-18 architecture with output dimension 512, followed by a four-layer MLP, also with feature and output dimensions of 512.

**Conclusion:** As observed in Table 5, PGR can flexibly condition on episodic curiosity to obtain strong performance in environments with highly stochastic transitions. Crucially, the generative replay PGR variants consistently outperform their model-free PPO counterparts which directly add exploration bonuses.

## B  COMBINING EXPLORATION BONUSES AND PGR

Our core contributions, a framework for prioritized generative replay and an exposition into why one particular instantiation works, are orthogonal to exploration. However, in Appendix A and also Section 5.1 of the main text, we show that PGR always outperforms exploration bonus variants. This raises an interesting question: how might exploration bonuses be combined with PGR?

|  | Quadruped-Walk | Cheetah-Run |
|---|---|---|
| REDQ | 496.75 ± 151.00 | 606.86 ± 99.77 |
| SYNTHER | 727.01 ± 86.66 | 729.35 ± 49.59 |
| REDQ + CURIOSITY | 687.14 ± 93.12 | 682.64 ± 52.89 |
| SYNTHER + CURIOSITY | 803.87 ± 41.52 | 743.39 ± 47.60 |
| PGR (CURIOSITY) | **927.98 ± 25.18** | **817.36 ± 35.93** |

Table 6: **Explicit Exploration Bonuses.** Results are on state-based DMC-100K (3 seeds, 1 std. dev. err.).

|  | Quadruped-Walk | Cheetah-Run |
|---|---|---|
| NOISYNETS | 688.29 ± 65.55 | 770.14 ± 70.53 |
| BOOT-DQN | 721.49 ± 39.82 | 754.56 ± 67.38 |
| PGR (NOISYNETS) | **939.32 ± 36.74** | 893.67 ± 43.47 |
| PGR (BOOT-DQN) | 903.29 ± 40.50 | **912.65 ± 47.22** |
| PGR (CURIOSITY) | 927.98 ± 25.18 | 817.36 ± 35.93 |

Table 7: **Implicit Exploration Bonuses.** Results are on state-based DMC-100K (3 seeds, 1 std. dev. err.).

### B.1  EXPLICIT EXPLORATION BONUSES.

We reiterate our results in Section 5.1 which demonstrate that PGR outperforms baselines augmented with an exploration bonus. Specifically, we examine the model-free baseline (REDQ), as well as the unconditional generative replay baseline (SYNTHER), when either is given an intrinsic curiosity bonus, learned via the same intrinsic curiosity module ($c.f.$ Eq. (5) and Pathak et al. (2017)). We follow the hyperparameters of Pathak et al. (2017) and set the intrinsic reward weight to 0.1. Results in Table 6 show that PGR continues to enjoy a healthy lead over such explicit exploration bonuses.

### B.2  IMPLICIT EXPLORATION BONUSES.

Next, we study how exploration bonuses can be involved more implicitly. In particular, we consider modifying the model class of the underlying policy to better facilitate exploration. We choose two well-established methods along this line of work — noisy networks (Fortunato et al., 2018) and bootstrapped Q-value exploration (Osband et al., 2016).

**Noisy Networks.** As a baseline, we replace the Q networks used in our REDQ model-free baseline with noisy Q networks from Fortunato et al. (2018). That is, each linear layer is transformed from:

$$y = wx + b, \tag{9}$$

for output $y$, and learnable weights and biases $w$ and $b$, respectively, to a "noisier" version:

$$y = (\mu^w + \sigma^w \odot \varepsilon^w)x + \mu^b + \sigma^b \odot \varepsilon^b. \tag{10}$$

Here, the noise parameters $\mu$ and $\sigma$ are learnable, whereas $\varepsilon$ are sampled noise random variables. Initialization of $\mu$, $\sigma$ and $\varepsilon$ all follow the recommendations in Section 3.2 of Fortunato et al. (2018) exactly.

**Bootstrapped Q-Values.** We also compare to the work of Osband et al. (2016), where uncertainty is injected via a bootstrapping mechanism across multiple $Q$-networks. In particular, suppose we have an ensemble of $K$ $Q$-networks, Given an episode, we select a single $Q$-value network to act through, and sample a binary mask $m$ of length $K$ according to some distribution $\mathcal{M}$. This mask describes the subset of $Q$-networks which are updated using this transition. This promotes exploration by preserving a temporally consistent estimate of uncertainty independently for each of $Q_1, \ldots, Q_K$.

This setup fits naturally with the $Q$-ensemble used in our REDQ baseline and can thus be readily integrated. We follow Osband et al. (2016) and simply use an all-one mask (i.e. masking distribution $\mathcal{M}$ of Bernoulli(1)), which corresponds to a basic ensemble of all $Q$ heads.

**PGR Variants.** To combine PGR with NOISYNETS, we only replace linear layers used in the policy (i.e. the $Q$-networks involved in the REDQ baseline.) That is, for the sake of minimal comparison, the embedding layers within the intrinsic curiosity-based relevance function are not modified. To combine PGR with BOOT-DQN, we make a similar choice.

**Conclusion:** Taken together, Table 6 and Table 7 demonstrate that PGR is empirically orthogonal to, and more useful than, simple exploration bonuses. In particular, explicit exploration bonuses do not allow baselines to outperform PGR, even when PGR is using the exact same bonuses to parameterize a relevance function. But implicit exploration bonuses can be readily integrated with the PGR framework, and may indeed bring complementary benefits, as we observe in the cheetah-run environment of Table 7. However, neither DMC-100K nor OpenAI Gym environments are well-suited to investigate exploration problems, and hence we avoid making any claims in the main text regarding exploration. We leave this intersection open as an interesting area for future work.

## C  MODEL-BASED RL BASELINES WITH NOISY DYNAMICS

In this section, we show that PGR is substantially different from model-based RL algorithms which generate trajectories through a learned dynamics model.

Our central argument is that under imperfect or noisy observations, PGR is more robust than methods which generate trajectories or transitions directly via a learned dynamics model. To verify this claim, we evaluate PGR against two such model-based baselines: MAX (Shyam et al., 2019) and DREAMER-V3 (Hafner et al., 2023). Crucially, we noise the observations within the training data to each model's learned dynamics components. For PGR, we add isotropic Gaussian noise to the current state $s$ and the next state $s'$ of $20\%$ of the transitions within each batch to the ICM. For MAX, we similarly modify $20\%$ of the transitions within the exploration data for training the model ensemble. For fair comparison, we also swap out the SAC backbone in MAX in favor of the more performant REDQ policy. Finally, for DREAMER-V3, we pollute states in transitions given to the recurrent state-space world model with the exact same noise and at the exact same rate as for PGR.

Practically, we found it much easier to convert MAX to the DMC benchmark, rather than convert DREAMER-V3 to the OpenAI Gymnasium benchmark that MAX originally presented results on. We choose a subset of the DMC tasks which correspond most closely with their OpenAI Gym counterparts, arriving at the cheetah-run, walker-walk and hopper-hop tasks presented in Table 8. Following our setup details in Section 5, we again allow for 100K online environment interactions.

|          |                  | Cheetah-Run        | Walker-Walk        | Hopper-Hop         |
|----------|------------------|--------------------|--------------------|--------------------|
| Original | MAX              | 644.79 ± 63.90     | 509.63 ± 48.76     | 48.30 ± 16.56      |
|          | DREAMER-V3       | 362.01 ± 30.69     | 627.79 ± 41.53     | 86.31 ± 21.21      |
|          | PGR (CURIOSITY)  | **817.36 ± 35.93** | **865.33 ± 75.39** | **94.45 ± 12.07**  |
| Noised   | MAX              | 363.26 ± 58.86     | 421.06 ± 33.77     | 17.07 ± 9.82       |
|          | DREAMER-V3       | 199.74 ± 34.60     | 386.13 ± 59.71     | 35.42 ± 18.03      |
|          | PGR (CURIOSITY)  | **697.67 ± 29.94** | **734.02 ± 48.57** | **64.45 ± 10.86**  |

Table 8: **Average returns on original and noisy state-based DMC-100K (5 seeds, 1 std. dev. err.).** We deliberately noise the actions seen during training by the dynamics models for various algorithms. PGR demonstrates more robustness to imperfect dynamics models.

As we observe from Table 8, PGR significantly outperforms both model-based algorithms which plan/generate trajectories *directly through* their learned dynamics. While such algorithms deeply couple the learning of environment dynamics and the policy, PGR offers an alternative and weaker coupling. In particular, PGR uses the ICM to learn transition probabilities $p(s'|s, a)$ (i.e. environment dynamics). The prediction error $e$ from the ICM then in turn serves as a conditioning signal to a diffusion model which learns $p(s, a, s', r \mid e)$. Thus, **even if the ICM is poorly trained due to imperfect input states, the diffusion model will still generate transition tuples $(\mathbf{s}, \mathbf{a}, \mathbf{s}', \mathbf{r})$ which adhere well to the true environment dynamics**.

**Conclusion:** PGR offers a distinct design choice from model-based RL algorithms which plan directly through a generative model. We empirically validate that this distinct design choice can be advantageous when dynamics predictions are imperfect due to noisy observations.

## D MISCELLANEOUS QUANTITATIVE EVALUATION

For comparison, Table 9 presents quantitative readouts of the PER baselines initially shown in Fig. 3a.

We also address questions regarding how PGR might be compatible with different generative model classes. In particular, we replace the underlying diffusion model in SYNTHER and PGR with an unconditional and conditional VAE, respectively. We design our VAE to approximate the capacity of the diffusion model in both SYNTHER and PGR (~6.8 million parameters). Specifically, we use a residual net encoder and decoder with 4 and 8 layers, respectively. Each layer has bottleneck dimension 128, with a latent dimension of 32, resulting in 2.6M encoder and 5.3M decoder parameters.

Results on three tasks (state-based DMC-100k) are shown in Table 9. We conclude that a) indeed conditional generation continues to outperform unconditional generation, but b) overall performance is much lower than either SYNTHER or PGR, and arguably within variance of the REDQ baseline. We believe this performance gap well justifies the focused use of diffusion models.

|  | Quadruped-Walk | Cheetah-Run | Reacher-Hard |
| --- | --- | --- | --- |
| REDQ | 496.75 ± 151.00 | 606.86 ± 99.77 | 733.54 ± 79.66 |
| SYNTHER | 727.01 ± 86.66 | 729.35 ± 49.59 | 838.60 ± 131.15 |
| PER (TD-ERROR) | 694.02 ± 99.17 | 685.23 ± 63.76 | 810.37 ± 89.22 |
| PER (CURIOSITY) | 726.93 ± 71.59 | 627.68 ± 55.50 | 763.21 ± 52.29 |
| UNCOND. VAE | 384.38 ± 154.30 | 549.36 ± 190.79 | 700.65 ± 161.18 |
| COND. VAE | 501.99 ± 79.88 | 668.49 ± 76.81 | 792.85 ± 93.73 |
| PGR (Curiosity) | **927.98 ± 25.18** | **817.36 ± 35.93** | **915.21 ± 48.24** |

Table 9: **Further results on DMC-100K.** We include prioritized experience replay baselines from Figure 2a of the main text for easier comparison. We also show PGR with different generative model classes, such as variational autoencoders (VAEs).

Finally, we show how we chose the frequency of the inner loop in Algorithm 1.

In Fig. 9, we plot the performance of PGR on the hopper-stand environment as a function of the frequency of the inner loop. We show that performance increases modestly the more frequently the inner loop is performed. This is intuitive — grounding the diffusion model more regularly in the real transitions allows it to better generate on-policy transitions, improving the stability of policy learning.

To tradeoff effectively between diminishing returns and training time, we use a heuristic elbow method, arriving at regenerating the diffusion buffer once every 10K iterations as "optimal."

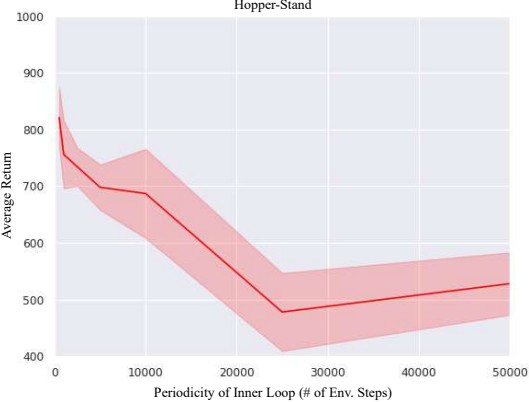

Figure 9: **PGR improves with increased frequency of the generative inner loop in Algorithm 1.** To tradeoff efficiency and performance, we select the inner loop to run periodically every 10K iterations.

