# OpenReview forum: "Prioritized Generative Replay"
_ICLR.cc/2025/Conference — ICLR 2025 Oral_

### Official Review · Reviewer_8yrp · 2024-10-31

**Soundness:** 4
**Presentation:** 3
**Contribution:** 3
**Rating:** 8
**Confidence:** 4

**Summary:**

This paper introduces a framework called Prioritized Generative Replay (PGR), a novel approach to enhance sample efficiency in online reinforcement learning (RL). Traditionally, replay buffers store experienced transitions and replay them uniformly or with prioritization based on metrics like TD-error. However, the authors point out that uniform replay can be inefficient, and prioritization can lead to overfitting. PGR addresses these issues by using a conditional generative model to create a parametric replay buffer.

The paper claims that this allows for two key advantages:
1) Densification: The generative model can create new, plausible transitions beyond those directly experienced, enriching the training data, especially in sparsely explored regions.
2) Guidance: By conditioning the generative model on "relevance functions," the generated transitions can be steered towards areas more critical for learning, such as states with high novelty or uncertainty.

The authors also explore various relevance functions, including return, TD-error, and curiosity. They find that curiosity, based on the prediction error of a learned dynamics model, performs best. This is attributed to its ability to promote diversity in the generated transitions, thus reducing overfitting. They also show that their approach consistently improves performance and sample efficiency in both state- and pixel based domains.

**Strengths:**

1) PGR offers a fresh perspective on replay buffers by combining generative modeling with guided replay. Framing this problem as a conditional generation problem with diffusion models is novel.
2) Diffusion model typically uses one single set of HPs requires no additional tuning I'd assume. This works well for PGR
3) Empirical results on various benchmarks demonstrate that PGR consistently outperforms existing model-free and model-based RL algorithms, as well as a generative replay baseline without guidance. Also has been shown to work in both state-based and pixel-based environments.
4) PGR is shown to scale well with larger policy networks and higher synthetic-to-real data ratios (important ablation that I wanted to see), potentially enabling more data-efficient training of large-scale RL agents. Really important result for scaling to many real use cases.
5) The authors also provide insights into why PGR works, particularly highlighting the role of curiosity in promoting diversity and reducing overfitting.

**Weaknesses:**

1) The curiosity-based relevance function relies on a learned dynamics model, which might be challenging to train accurately in complex environments.
2) Increasing Synthetic Data ratio does not benefit PGR and the unconditional baseline (SynthER) equally. PGR scales better at r=0.75 than SYNTHER but neither benefits from 0.875. We would think the trend would be consistent? whats the intution behind this? Also this figure 7 could be improved with the variation in r being shown
3) (Minor) writing issues throughout the paper with some missing words etc. Please re-read the paper and make the necessary changes.

**Questions:**

1) How robust is PGR to errors in the learned dynamics model? Are there ways to mitigate the impact of inaccurate dynamics predictions on the curiosity-based relevance function?
2) Could PGR be extended to offline RL settings? If so, what modifications would be necessary?
3) How does PGR's performance compare against PER baselines which use approximate parametric models of prior experience?
4) Are there any other relevance functions thats been tried out? As thats core to the working of PGR.

---

> ### Author Response · Authors · 2024-11-19
> **Response to reviewer 8yrp (1/2)**
>
> Thank you very much for your review.
>
> **Q1:** “How robust is PGR to errors in the learned dynamics model? Are there ways to mitigate the impact of inaccurate dynamics predictions on the curiosity-based relevance function?”
>
> **A1:** **Like most works in model-based RL, PGR is indeed susceptible to errors in the learned dynamics model of the curiosity-based relevance function. However, in practice, we show that PGR is far more robust than other approaches, especially those which directly plan through the dynamics model.**
>
> Specifically, we perform an experiment comparing MAX[1] and Dreamer-v3[4] to PGR when the learned dynamics are misled by noisy observations. We inject Gaussian noise into some fraction of the observations in transitions fed to the world model in Dreamer-v3, as well as the observations in transitions fed to the exploration model ensemble in MAX. We similarly pollute the observations in transitions to the ICM module in PGR. For more details, we refer the reviewer to Appendix C. We reproduce results here for ease of viewing:
>
> |          |                 | Cheetah-Run            | Walker-Walk            | Hopper-Hop            |
> |:----------:|-----------------|------------------------|------------------------|-----------------------|
> | &#8593; | MAX             | 644.79 $\pm$ 63.90     | 509.63 $\pm$ 48.76     | 48.30 $\pm$ 16.56     |
> | Original | Dreamer-v3      | 362.01 $\pm$ 30.69     | 627.79 $\pm$ 41.53     | 86.31 $\pm$ 21.21     |
> | &#8595; | PGR (Curiosity) | **817.36 $\pm$ 35.93** | **865.33 $\pm$ 75.39** | **94.45 $\pm$ 12.07** |
> | &#8593; | MAX             | 363.26 $\pm$ 58.86     | 421.06 $\pm$ 33.77     | 17.07 $\pm$ 9.82      |
> | Noised   | Dreamer-v3      | 199.74 $\pm$ 34.60     | 386.13 $\pm$ 59.71     | 35.42 $\pm$ 18.03     |
> | &#8595; | PGR (Curiosity) | **697.67 $\pm$ 29.94** | **734.02 $\pm$ 48.57** | **64.45 $\pm$ 10.86** |
>
> As we observe, PGR continues to outperform MAX and Dreamer-v3 under these noisier conditions. This is because Dreamer must learn the joint distribution over $p(s, a, s’, r)$ entirely, and thus any partial model errors propagate throughout generations, and also affect policy learning. In general, (with or without intrinsic reward,) planning _directly through_ a world model with inaccurate dynamics will naturally lead to poor performance. Contrast this with PGR, which more loosely couples the learning of environment dynamics (via intrinsic curiosity) from the rest of the generative world model (diffusion over $(s, a, s’, r)$ tuples). Concretely, PGR learns the dynamics $p(s’ | s, a)$, and conditions on dynamics error $e$ to generate $(s, a, s’, r | e)$. This means that even with an imperfect dynamics model, we may obtain a poor signal through $e$, but the generated transitions $(s, a, s’, r)$ still remain faithful to the ground truth environment dynamics! Furthermore, densification means PGR also grounds synthetic data with real data, allowing the policy to be more robust to any erroneous synthetic generations.
>
> More generally, one way to mitigate the impact of inaccurate dynamics predictions for PGR is to leverage a more robust relevance function $\mathcal{F}$. For example, one could use a variational approach to intrinsic rewards such as VIME [4], with better calibrated uncertainty, to either a) directly do conditional generation of more robust transitions, or b) to downweigh generated samples when the dynamics are highly uncertain.
>
> ---
>
> **Q2:** “PGR scales better at r=0.75 than SynthER but neither benefits from 0.875. We would think the trend would be consistent? What is the intuition behind this?”
>
> **A2:** **There is nothing better than ground truth data directly obtained in the “real world”. While generative replay is a useful mechanism to densify past real-world online experience, further work is needed to completely replace real environment interactions with synthetic ones.**
>
> We show in Figure 5 of the main text that our diffusion model is a reasonable approximation to the simulator dynamics, but this MSE is still non-zero. When the synthetic data ratio is too high, it is plausible that the model begins fitting to the biased distribution of the diffusion model. Thus, overly relying on synthetic data may result in policy collapse.
>
> Note that we did not tune the hyperparameter controlling the frequency of the inner loop when varying the synthetic data ratio $r$. We hypothesize that 10K iterations per inner loop might no longer be optimal for higher synthetic data ratios, since more regularly grounding the diffusion model in the real simulator dynamics becomes more critical if we rely more heavily on the generated data for policy learning.

---

> > ### Author Response · Authors · 2024-11-19
> > **Response to reviewer 8yrp (2/2)**
> >
> > **Q3:** “Could PGR be extended to offline RL settings? If so, what modifications would be necessary?”
> >
> > **A3:** **While parts of our method are definitely applicable to the offline setting (e.g. the idea of using conditional signals to guide generation towards pre-specified data characteristics), our problem setting is strictly in online RL.**
> >
> > We believe this choice makes our contribution clearer — namely, we show how widely-used prioritization mechanisms in online RL can be cast through the lens of conditional generation. In online RL, data is expensive to obtain (motivating generative densification) and relevant data is scarce (motivating generative guidance). Both these elements are directly addressed and integrated into the fabric of PGR.
> >
> > We leave to future work to investigate how our insights and techniques can be further leveraged to the offline RL setting. One important modification is that with entire offline trajectories available, it may make sense to use relevance functions with an understanding of trajectory-level semantics, for improved guidance during densification.
> >
> > ---
> >
> > **Q4:** “How does PGR's performance compare against PER baselines which use approximate parametric models of prior experience?”
> >
> > **A4:** **PGR outperforms baselines which use parametric models of prior experience.**
> >
> > We direct the reviewer to Figure 3a, and the accompanying discussion on lines 369-374 in Section 5.1. We compare PGR to two PER baselines: one which uses the learned Q-networks to estimate TD-error for prioritization, and the other which uses the learned ICM module to prioritize transitions based on intrinsic curiosity. This latter module has the exact same parameterization as the relevance function in the curiosity variant of PGR. As we observe in Figure 3a, PGR outperforms these baselines. For ease of quantitative comparison, we display these values in Table 9 of Appendix D, and reproduce the results here:
> >
> > |                 | Quadruped-Walk         | Cheetah-Run            | Reacher-Hard           |
> > |-----------------|------------------------|------------------------|------------------------|
> > | REDQ            | 496.75 $\pm$ 151.00    | 606.86 $\pm$ 99.77     | 733.54 $\pm$ 79.66     |
> > | SynthER         | 727.01 $\pm$ 86.66     | 729.35 $\pm$ 49.59     | 838.60 $\pm$ 131.15    |
> > | PER (TD-Error)  | 694.02 $\pm$ 99.17     | 685.23 $\pm$ 63.76     | 810.37 $\pm$ 89.22     |
> > | PER (Curiosity) | 726.93 $\pm$ 71.59     | 627.68 $\pm$ 55.50     | 763.21 $\pm$ 52.29     |
> > | PGR (Curiosity) | **927.98 $\pm$ 25.18** | **817.36 $\pm$ 35.93** | **915.21 $\pm$ 48.24** |
> >
> > ---
> >
> > **Q5:** “Are there any other relevance functions that have been tried out?”
> >
> > **A5:** In the main text, we present results using relevance functions defined via raw reward, value estimation, TD-error and intrinsic curiosity. Following the discussion period, we include additional results for conditioning on alternative intrinsic rewards such as RND [1] and pseudo-counts [2], as well as online episodic curiosity [3] for harder environments where stochasticity impedes naive prediction error-based curiosity. For a full discussion and accompanying results, we refer the reviewer to the above general response, as well as Tables 4 and 5 in Appendix A.
> >
> > ---
> >
> > **Q6:** “Minor writing issues with some missing words”
> >
> > **A6:** We have made minor edits throughout to improve the readability of the paper. Thank you for the detailed read!
> >
> >
> > [1] Burda et al. “Exploration by Random Network Distillation.” arXiv 2018.
> >
> > [2] Bellemare et al. “Unifying Count-Based Exploration and Intrinsic Motivation.” NeurIPS 2016.
> >
> > [3] Savinov et al. “Episodic Curiosity Through Reachability.” arXiv 2018.
> >
> > [4] Houthooft et al. “VIME: Variational Information Maximizing Exploration” NeurIPS 2016.

---

> > > ### Comment · Reviewer_8yrp · 2024-11-25
> > >
> > > Thank you for providing detailed answers to my questions and adding in some extra details - such as relevance functions etc.
> > > This gives me much more clarity of the work and I believe this would be a good addition to the community. I have raised my score to reflect my new opinion.

---

> ### Author Response · Authors · 2024-11-25
> **Following Up on Discussion Period**
>
> Dear reviewer 8yrp,
>
> Thank you for the constructive feedback and suggestions. We greatly appreciate you updating your score. If there are any additional clarifications we can provide, please let us know. Thank you once again.

---

### Official Review · Reviewer_ir9Q · 2024-11-04

**Soundness:** 3
**Presentation:** 3
**Contribution:** 3
**Rating:** 6
**Confidence:** 3

**Summary:**

The paper proposes to use conditional diffusion models to improve the experience replay for an RL learning agent. The method proposed improves performance by improving the diversity of the samples in the experience replay buffer and reducing overfitting.

**Strengths:**

1. The paper is well written and provides a clear explanation of their method.
2. The research problem addressed in the paper is well laid out and is an important one to improve the performance of RL methods.

**Weaknesses:**

1. While the method shows improved performance, it is a bit simple as it combines existing elements in diffusion models and RL to propose the solution.
2. It would be useful to compare the effect of different kinds of exploration bonuses.

**Questions:**

1. Is the method compatible with different kinds of exploration bonuses? If so, how do you think they would compare?
2. How do you think the method would do when simply having diverse samples does not imply usefulness? An example is the noisy tv problem.
3. How sensitive is the algo towards the frequency of the inner loop in Algo 1?
4. Can multiple relevance functions be combined?

---

> ### Author Response · Authors · 2024-11-19
> **Response to reviewer ir9Q (1/2)**
>
> Thank you very much for your review.
>
> **Q1:** “While the method shows improved performance, it is a bit simple as it combines existing elements in diffusion models and RL to propose the solution.”
>
> **A1:** **To the best of our knowledge, our work is the first to cast widely-used prioritized replay in online RL through the lens of conditional generation. Moreover, we provide a novel and central role to curiosity, as a useful prioritization signal during conditional data generation. This is a novel use case well outside its usual stage in RL methods, which have historically only leveraged it as intrinsic reward for better exploration.**
>
> The novelty of our approach does not lie in using a diffusion model (conditional or unconditional) to capture the replay buffer. As we emphasize through lines 410-432 in Section 5.2 of the main text, knowing what to condition on is actually more important than choosing to do conditioning (or generation at all) in the first place. Moreover, why and how conditioning works are also equally valuable questions to answer, and we provide generalizable and non-obvious insights into both these elements.
>
> ---
>
> **Q2:** “Is the method compatible with different kinds of exploration bonuses? If so, how do you think they would compare?”
>
> **A2:** **We show in our general response that the contributions in PGR go beyond explicit exploration bonuses such as intrinsic curiosity or RND [1], or methods which implicitly promote exploration such as bootstrapped DQN [4] or NoisyNets [5].**
>
> These latter methods are readily compatible with PGR, requiring only light modifications to the underlying baseline REDQ architecture. We reproduce the results in this comparison below (experimental details can be found in Appendix B):
>
> |                 | Quadruped-Walk         | Cheetah-Run            |
> |-----------------|------------------------|------------------------|
> | NoisyNets       | 688.29 $\pm$ 65.55     | 770.14 $\pm$ 70.53     |
> | Boot-DQN        | 721.49 $\pm$ 39.82     | 754.56 $\pm$ 67.38     |
> | PGR (NoisyNets) | **939.32 $\pm$ 36.74** | 893.67 $\pm$ 43.47     |
> | PGR (Boot-DQN)  | 903.29 $\pm$ 40.50     | **912.65 $\pm$ 47.22** |
> | PGR (Curiosity) | 927.98 $\pm$ 25.18     | 817.36 $\pm$ 35.93     |
>
> Our results indicate densification via PGR is orthogonal to and more useful than promoting exploration alone. For a more comprehensive discussion, we direct the reviewer to our general response above.
>
> ---
>
> **Q3:** “How do you think the method would do when simply having diverse samples does not imply usefulness? An example is the noisy tv problem.”
>
> **A3:** **We also show in our general response that PGR can effectively condition on relevance functions based on RND [1], pseudo-counts [2] or episodic curiosity [3], which all demonstrated reliably evading stochastic noise sources in prior work.**
>
> The relevant discussion can be found under the first question in the above general response, with experimental details in Appendix A. We also direct the reviewer to our response to reviewer 1k48. Specifically, we argue that the search space for desirable relevance functions $\mathcal{F}$ is not that large, and that prediction error-based intrinsic curiosity (ICM) serves as a good default choice. In the event that ICM is insufficient due to stochastic noise sources such as noisy TVs, this problem has been shown in [1] and [3] to be easy to diagnose, and we demonstrate in our general response above that swapping in a more appropriate $\mathcal{F}$ restores the benefits of PGR.
>
> Finally, we remark that we do not generate diverse samples _because_ they imply usefulness. Rather, our first-order goal is to generate samples that are relevant and useful (e.g. reduce early overfitting of the Q-function.) And as a possible mechanistic explanation for this property, we empirically show that these samples are actually more diverse than those generated via the unconditional baseline.

---

> > ### Author Response · Authors · 2024-11-19
> > **Response to reviewer ir9Q (2/2)**
> >
> > **Q4:** “How sensitive is the algo towards the frequency of the inner loop in Algo 1?”
> >
> > **A4:** **PGR benefits from a more frequent inner loop in Algo 1, but the tradeoff with training time is sublinear. In our work, we identify via a simple elbow validation curve that a frequency of once every 10K iterations is optimal.**
> >
> > In Figure 9 of Appendix D, we plot the performance of PGR on the hopper-stand environment as a function of the frequency of the inner loop. As mentioned in Section 5.1 of the main text, this is the environment upon which we tuned the hyperparameters for PGR. We show that performance increases modestly the more frequently the inner loop is performed. This is intuitive — grounding the diffusion model more regularly in the real transitions allows it to better generate on-policy transitions, improving the stability of policy learning.
> >
> > To tradeoff effectively between diminishing returns and training time, we use a heuristic elbow method on Figure 9, concluding that performing the inner loop once every 10K iterations is “optimal.”
> >
> > ---
> >
> > **Q5:** “Can multiple relevance functions be combined?”
> >
> > **A5:** **Multiple relevance functions can indeed be combined. While this is not a focal point of our work, we highlight a few ways to do so here.**
> >
> > Relevance functions that provide complementary benefits can be easily combined. One way to do so is to sum or concatenate the function embeddings, after appropriate normalization. Then, passing the new scalar or vector embedding to the existing framework blithely is sufficient. Another reasonable modification is to allow separate conditioning on either or both function embedding within the diffusion model. Note this would involve more careful tuning of the CFG coefficients corresponding to each relevance function condition.
> >
> > In practice, complementary relevance functions in the online setting are non-obvious to come up with. As we show in our work, PGR with curiosity is unanimously superior to reward, value or TD-error based relevance functions. We believe that this line of research is more relevant for the offline setting, where large-scale pretrained priors with complementary properties can be used for guided generation. However, if the reviewer feels that this discussion is important to include in the main text, we are happy to add an additional section within the appendix.
> >
> > [1] Burda et al. “Exploration by Random Network Distillation.” arXiv 2018.
> >
> > [2] Bellemare et al. “Unifying Count-Based Exploration and Intrinsic Motivation.” NeurIPS 2016.
> >
> > [3] Savinov et al. “Episodic Curiosity Through Reachability.” arXiv 2018.
> >
> > [4] Osband et al. “Deep Exploration via Bootstrapped DQN.” NeurIPS 2016.
> >
> > [5] Fortunato et al. “Noisy Networks for Exploration.” ICLR 2018.

---

> > > ### Author Response · Authors · 2024-11-25
> > > **Following Up on  Discussion Period**
> > >
> > > Dear reviewer ir9Q,
> > >
> > > Thank you for the constructive feedback and suggestions. As the discussion period is coming to a close, we would appreciate you kindly reading our response. If we have addressed your questions and alleviated your concerns, we would also appreciate if you would consider updating your score.
> > >
> > > Please let us know if there are other additional clarifications we can provide. Thank you once again.

---

> > > > ### Comment · Reviewer_ir9Q · 2024-11-26
> > > >
> > > > I would like to thank the authors for putting in all the work in this rebuttal and for clarifying my questions as well as the questions of the other reviewers. Their efforts have been helpful in improving my understanding of their work and I am updating my score to reflect this.

---

### Official Review · Reviewer_Yyu8 · 2024-11-04

**Soundness:** 4
**Presentation:** 3
**Contribution:** 4
**Rating:** 8
**Confidence:** 3

**Summary:**

This paper proposes a conditional diffusion model as a synthetic replay buffer to combat early overfitting in online reinforcement learning and to diversify experiences for the learning model. This is achieved with a relevance function that selects samples that are rare but important for learning based on the temporal difference error, the value of a learned Q-function and a modified intrinsic curiosity module.

**Strengths:**

One of the strength of this paper are the clear and concise language as well as good structured presentation of the proposed method.
It is quite logical to improve on the already existing prioritized experience replay method and implement it in the generative domain. The method is explained well and should be quite easily reproducable.
Overall the research could be a valuable contribution to the reinforcement learning community.

**Weaknesses:**

A topic i feel like missed somewhat are the different ways to approach generative replay such as mentions of other generative models (e.g. variational auto encoders, gaussian mixture models) and why they were not used.
One thing i found rather off putting and this is very nitpicky is that the Tables 1, 2 and 3 are a bit crammed and slightly off from each other.

**Questions:**

What exploration method does the agent use?
Could the exploration method be improved instead of the sample generation to improve diversity of samples?
Would a combination of both a better exploration and this method be the optimal and a possible solution?

---

> ### Author Response · Authors · 2024-11-19
> **Response to reviewer Yyu8**
>
> Thank you very much for your review.
>
> **Q1:** “Discussing different ways to approach generative replay such as mentions of other generative models (e.g. variational auto encoders, gaussian mixture models) and why they were not used.”
>
> **A1:** **We believe that VAEs and GANs have issues which make comparison to baselines harder and would only obfuscate our contribution.** In particular, GANs are sensitive to hyperparameters and much harder to train [1]. VAEs do not have the generation quality that is required to achieve comparable results to methods like SynthER. A baseline level of generation quality is especially important in the online setting, when initial data is noisy.
>
> To verify this, we replace the underlying diffusion model in SynthER and PGR with an unconditional and conditional VAE, respectively. We design our VAE to approximate the capacity of the diffusion model in both SynthER and PGR (~6.8 million parameters). Specifically, we use a ResNet-style encoder and decoder with 4 and 8 layers, respectively. Each residual layer has bottleneck dimension 128, with a latent dimension of 32, resulting in 2.6 million encoder and 5.3 million decoder parameters. Note that this model architecture is also highly similar to the residual MLP denoiser used in the diffusion models of SynthER and PGR.
> Results on three tasks (state-based DMC-100k) are shown in Appendix D, with results reproduced below for ease of viewing:
>
> |                   | Quadruped-Walk         | Cheetah-Run            | Reacher-Hard           |
> |-------------------|------------------------|------------------------|------------------------|
> | REDQ              | 496.75 $\pm$ 151.00    | 606.86 $\pm$ 99.77     | 733.54 $\pm$ 79.66     |
> | SynthER           | 727.01 $\pm$ 86.66     | 729.35 $\pm$ 49.59     | 838.60 $\pm$ 131.15    |
> | Unconditional VAE | 384.38 $\pm$ 154.30    | 549.36 $\pm$ 190.79    | 700.65 $\pm$ 161.18    |
> | Conditional VAE   | 501.99 $\pm$ 79.88     | 668.49 $\pm$ 76.81     | 792.85 $\pm$ 93.73     |
> | PGR (Curiosity)   | **927.98 $\pm$ 25.18** | **817.36 $\pm$ 35.93** | **915.21 $\pm$ 48.24** |
>
> We conclude that a) indeed conditional generation continues to outperform unconditional generation, but b) overall performance is much lower than either SynthER or PGR, arguably performing within variance of the REDQ baseline. We believe this performance gap well justifies the focused use of diffusion models.
>
> ---
>
> **Q2:** “Formatting of Tables 1, 2, and 3.”
>
> **A2:** We have reformatted the paper to better accommodate Tables 1, 2 and 3. Thank you for the detailed comment!
>
> ---
>
> **Q3:** “What exploration method does the agent use? Could the exploration method be improved instead of the sample generation to improve diversity of samples? Would a combination of both a better exploration and this method be the optimal and a possible solution?”
>
> **A3:** **PGR does not rely on any particular exploration method. Indeed, better exploration can be orthogonally combined with PGR as a plausible approach.**
>
> Note that while better exploration alone may improve the diversity of samples, we show in Appendix B that the densification offered via generative replay is independently and critically important to PGR’s improved policy learning results.
>
> As we illustrate in our general response above, PGR offers distinct advantages over explicit exploration bonuses, and potentially can be combined with implicit exploration bonuses. While we do not look at exploration in this work, we believe that these initial findings have now opened up an interesting intersection between generative replay and exploration for future works in online RL.
>
>
> [1] Salimans et al. “Improved Techniques for Training GANs.” arXiv 2016.

---

> > ### Author Response · Authors · 2024-11-25
> > **Following Up on Discussion Period**
> >
> > Dear reviewer Yyu8,
> >
> > Thank you for the constructive feedback and suggestions. As the discussion period is coming to a close, we would appreciate you kindly reading our response. If we have addressed your questions and alleviated your concerns, we would also appreciate if you would consider updating your score.
> >
> > Please let us know if there are other additional clarifications we can provide. Thank you once again.

---

> > > ### Comment · Reviewer_Yyu8 · 2024-11-25
> > >
> > > Dear Authors,
> > > sorry for the late response to your rebuttal. Thank you for your detailed answers to the questions I had. Your thorough rebuttal has alleviated all my concerns regarding your paper. I find it particularly interesting how VAEs and GANs perfomed within your context or rather didn't perform all too well. I personally think that better exploration strategies can be interesting and necessary future research to improve not only your approach but others as well.

---

> > > > ### Comment · Reviewer_1k48 · 2024-12-03
> > > >
> > > > Thank you for the substantive comments and discussion. I have raised my score accordingly.

---

### Official Review · Reviewer_1k48 · 2024-11-05

**Soundness:** 3
**Presentation:** 3
**Contribution:** 3
**Rating:** 8
**Confidence:** 4

**Summary:**

The work proposes a form of sample based experience replays that leverages a generative model to provide and augment samples drawn from the replay buffer. To avoid overfitting, a set of guidance functions are used to steer the generative process toward diverse and useful samples. The generative replay mechanism is a diffusion model that is conditioned on some auxiliary information. The authors propose a few different versions of this conditioning such as intrinsic curiosity, TD error, or Q values. The idea is that using these scores, the generative model can be steered towards generating high quality samples. Given such a replay mechanism, this work evaluates model free and model-based RL agents trained via this generative replay on gym and dmc.The results show improvement on both pixel based and state based tasks. There are also ablations with larger policy networks and higher generative data rations, which show further improvements.

-------------------------------------------------
I thank the authors for a substantive rebuttal that addressed my and (as far as I can tell) other concerns. I therefore raise my score to an 8.

**Strengths:**

* This work proposes a scalable method for training model-free or model-based agents in a variety of domains. I believe the formulation is simple enough to be integrated into and improve other approaches.

* I also found the presentation clear and easy to read.

* I found the scaling experiments to be very compelling, I'm a little concerned about the general thrust of driving up the syn-real data ratio as high as possible, since we do need to ground the generations in real experience. But I still think insights here are valuable.

**Weaknesses:**

I have two points of contention with this work.
1. From a paradigm perspective, I don't understand how this is different from prior work in model-based RL that apples intrinsic rewards to a learned dynamics model [1] or world-model [2]. These methods also utilize a generative model as a copy of the environment, then train the agent in simulation to acquire interesting data (under the intrinsic reward). It seems that this method does the same, except that instances, rather than full trajectories are generated. I do see how this is different than just applying an intrinsic bonus during training, since here the synthetic data has a chance to be more diverse.

2. I thank the authors for providing numerous experiments, but I am not at all convinced that this method is robust to the choice of guiding function F. ICM is known to be susceptible to the noisy TV problem, where difficult-to-model environmental factors score arbitrarily high under ICM. The chosen tasks are too simple perceptually to see this problem. This in and of itself is not a problem, but it means that we need to search for another F that works for our task which is hard in practice. In the meantime, there are other intrinsic rewards that do not suffer from this pathology [3].



[1] Shyam, Pranav, Wojciech Jaśkowski, and Faustino Gomez. "Model-based active exploration." International conference on machine learning. PMLR, 2019.

[2] Hafner, Danijar, et al. "Dream to control: Learning behaviors by latent imagination." arXiv preprint arXiv:1912.01603 (2019).

[3] Savinov, Nikolay, et al. "Episodic curiosity through reachability." arXiv preprint arXiv:1810.02274 (2018).

**Questions:**

I'll rephrase my above concerns as questions.

1. How is this method novel with respect to prior work that uses intrinsic rewards on rollouts from a learned dynamics model? It seems like a very similar approach to acquiring data that scores well under a given guidance function F, where F can be ICM or another intrinsic reward.

2. How does this method handle noisy-tvs?

---

> ### Author Response · Authors · 2024-11-19
> **Response to reviewer 1k48 (1/2)**
>
> Thank you very much for your review.
>
> **Q1:** How is this method novel with respect to prior work that uses intrinsic rewards on rollouts from a learned dynamics model?
>
> **A1:** **The PGR paradigm enables 1) generative _densification_ of online experience and 2) conditional _guidance_ via relevance functions. We show that each of these benefits uniquely positions our work amongst prior work.**
>
> Firstly, we note that densification is different from planning through the generative model entirely, as in e.g Dreamer [2][4]. In particular, Dreamer analytically backprops the gradients of dynamics through entire trajectories “generated” in latent space. This means that its generative world model is the bottleneck for learning. We show that in the presence of noisy/imperfect observations, approaches akin to this are less desirable than PGR.
>
> Specifically, we perform an experiment comparing MAX[1] and Dreamer-v3[4] to PGR when the learned dynamics are misled by noisy observations. We inject Gaussian noise into some fraction of the observations in transitions fed to the world model in Dreamer-v3, as well as the observations in transitions fed to the exploration model ensemble in MAX. We similarly pollute the observations in transitions to the ICM module in PGR. For more details, we refer the reviewer to Appendix C. We reproduce results here for ease of viewing:
>
> |          |                 | Cheetah-Run            | Walker-Walk            | Hopper-Hop            |
> |:----------:|-----------------|------------------------|------------------------|-----------------------|
> | &#8593; | MAX             | 644.79 $\pm$ 63.90     | 509.63 $\pm$ 48.76     | 48.30 $\pm$ 16.56     |
> | Original | Dreamer-v3      | 362.01 $\pm$ 30.69     | 627.79 $\pm$ 41.53     | 86.31 $\pm$ 21.21     |
> | &#8595; | PGR (Curiosity) | **817.36 $\pm$ 35.93** | **865.33 $\pm$ 75.39** | **94.45 $\pm$ 12.07** |
> | &#8593; | MAX             | 363.26 $\pm$ 58.86     | 421.06 $\pm$ 33.77     | 17.07 $\pm$ 9.82      |
> | Noised   | Dreamer-v3      | 199.74 $\pm$ 34.60     | 386.13 $\pm$ 59.71     | 35.42 $\pm$ 18.03     |
> | &#8595; | PGR (Curiosity) | **697.67 $\pm$ 29.94** | **734.02 $\pm$ 48.57** | **64.45 $\pm$ 10.86** |
>
> As we observe, PGR continues to outperform MAX and Dreamer-v3 under these noisier conditions. This is because Dreamer must learn the joint distribution over $p(s, a, s’, r)$ entirely, and thus any partial model errors propagate throughout generations, and also affect policy learning. In general, (with or without intrinsic reward,) planning _directly through_ a world model with inaccurate dynamics will naturally lead to poor performance. Contrast this with PGR, which more loosely couples the learning of environment dynamics (via intrinsic curiosity) from the rest of the generative world model (diffusion over $(s, a, s’, r)$ tuples). Concretely, PGR learns the dynamics $p(s’ | s, a)$, and conditions on dynamics error $e$ to generate $(s, a, s’, r | e)$. This means that even with an imperfect dynamics model, we may obtain a poor signal through $e$, but the generated transitions $(s, a, s’, r)$ still remain faithful to the ground truth environment dynamics! Furthermore, densification means PGR also grounds synthetic data with real data, allowing the policy to be more robust to any erroneous synthetic generations.
>
> Secondly, we are not concerned with intrinsic rewards for better exploration as in [1], which is often the use case for combining intrinsic rewards and model-based RL algorithms [1][5][6]. Using intrinsic reward to guide exploration is fundamentally different from using relevance functions to guide generation. In fact, PGR is not attached to using intrinsic rewards, at all. Our paradigm centers on _guidance_ for generative replay, meaning we can condition on **any** relevance function with desired properties. For example, we may choose $\mathcal{F}$ to explicitly avoid undesirable degenerate solutions. Concretely, we may have physics-based models which inform us apriori which $(s, a, s’, r)$ transition tuples are more likely to lead to unstable gaits for locomotion tasks, and penalize generating such transitions through an appropriate negative condition. Or, we might want to avoid the boundaries of a workspace and therefore decrease generations of $(s, a, s’, r)$ tuples so that $(s + a)$ is outside some epsilon bubble of the edges of the workspace.
>
> Due to time constraints we do not provide an empirical verification of these experiments. However, we believe that by showing an instantiation of PGR with ICM, and empirically grounding why and how it is superior to other simpler choices, our work opens up these promising new avenues for future research in online RL.
>
> In the meantime, we have softened the language in the paper which might erroneously provide the impression that ICM is the ideal, optimal or only choice for relevance functions.

---

> > ### Author Response · Authors · 2024-11-19
> > **Response to reviewer 1k48 (2/2)**
> >
> > **Q2:** “How does this method handle the noisy-tv [problem]?”
> >
> > **A2:** **We also show in our general response that PGR can effectively condition on relevance functions based on RND [7], pseudo-counts [8] or online episodic curiosity (ECO) [3], which all demonstrated reliably evading stochastic noise sources in prior work.**
> >
> > The relevant discussion can be found under the first question in the above general response, with experimental details in Appendix A in the updated draft. To summarize, PGR using RND or ECO as relevance functions continues to enjoy superior performance over baselines that do not perform generative replay.
> >
> > Our research contribution is _not_ the choice of relevance function. We merely use prediction error-based curiosity to illustrate a strong candidate with more desirable properties than previous obvious choices such as reward, value or TD-error, and provide an empirical explanation for why. That is, our core contribution centers on the value of this generative replay paradigm as a whole, particularly in online settings. Under this problem setting, data is expensive to obtain (motivating generative densification) and relevant data is scarce (motivating generative guidance). These are two issues fundamentally addressed by the PGR paradigm.
> >
> > **Finally, as the reviewer identifies, all this implies that we may need to search for an effective relevance function appropriate for our task and environment. However, we argue that this is not in practice an issue.**
> >
> > Firstly, we have shown that curiosity as defined via the ICM prediction error is a broadly effective default choice for PGR (in most standard online RL environments accepted by the wider community.) Even when ICM fails, as Savinov et al. [3] in Section 4.2 point out,  “visual inspection of the ICM learnt behaviour” makes it obvious to diagnose the noisy-TV problem, in which case we can opt for more robust relevance functions such as ECO. That is, we view the choice of relevance function as a hyperparameter, with curiosity à la ICM as a strong default. This is no different from the minimum reachable distance, task/bonus weights or episodic memory buffer hyperparameters described in Section 2.2 and 2.3 of Savinov et al. [3], which are requisite for the strong performance of ECO.
> >
> > Secondly, we feel that the majority of prior work investigating the noisy-TV problem with empirical results do so in artificial settings which do not reflect most settings where randomness is more structured. Concretely, Savinov et al. [3], Raileanu & Rocktäschel [9] and Burda et al. [7] (who first coined the noisy-TV term), all characterize this problem in a grid-like maze navigation setting, where often a literal noisy TV screen is playing random noise/images. In our particular online setting, that of continuous control with underlying physics-driven structure, this problem is very far removed. In general, we believe that in _most_ practical online settings, including the real world, stimuli may initially appear random to the learning agent, but such stochasticity is generally structured and exploitable for the vast majority of interesting tasks.
> >
> > [1] Shyam et al. “Model-based Active Exploration.” ICML 2019.
> >
> > [2] Hafner et al. “Dream to Control: Learning Behaviors by Latent Imagination.” arXiv 2019.
> >
> > [3] Savinov et al. “Episodic Curiosity Through Reachability.” arXiv 2018.
> >
> > [4] Hafner et al. “Mastering Diverse Domains through World Models.” arXiv 2023.
> >
> > [5] Mendoca et al. “Discovering and Achieving Goals via World Models.” NeurIPS 2021.
> >
> > [6] Sekar et al. “Planning to Explore via Self-Supervised World Models.” ICML 2020.
> >
> > [7] Burda et al. “Exploration by Random Network Distillation.” arXiv 2018.
> >
> > [8] Osband et al. “Deep Exploration via Bootstrapped DQN.” NeurIPS 2016.
> >
> > [9] Raileanu & Rocktäschel. “RIDE: Rewarding Impact-Driven Exploration for Procedurally-Generated Environments.” ICLR 2020.

---

> > ### Comment · Reviewer_1k48 · 2024-11-19
> > **Reply to Rebuttal**
> >
> > Thank you very much for your thorough rebuttal. I am going to consolidate my replies here.
> >
> > ## Overview
> >
> > I had two issues, one was the equivocation between model-based methods like MAX and Dreamer, and the other was the difficulty of choosing a relevance function. I think both concerns have been addressed, and I encourage the authors to add as many of these new results to the main paper as they can.
> >
> > Specifically:
> >
> > ## Noisy-TVs and Relevance Functions
> >
> > I'm ok with the assertion that noisy-tvs are a thought experiment -- and taken too literally as a criticism. I was mainly concerned with the difficulty of choosing an appropriate F. We may indeed have such a physics model, but I think in practice that is unlikely. Table 4 in the appendix shows some robustness to choice of exploration bonus, and I think that's good enough for me. I would like the author's take on table 5, it seems to suggest that this method does in fact suffer from this problem to some degree.
> >
> > ## Comparison to Model-based methods
> >
> > I appreciate the new table giving a comparison between Dreamer, MAX, and PGR. I still don't think there's much of a difference between "densification of the existing data distribution" and "improving the data distribution" via PGR and Exploration respectively. But I acknowledge that the looser dependance on grounded trajectories obtained via PGR vs a world-model makes a difference empirically.
> >
> > ---------------------------------------------------------
> >
> > Pending the rest of the rebuttal, I am inclined to raise my score to a 7.

---

> ### Author Response · Authors · 2024-11-20
> **Continuing Discussion with Reviewer 1k48 (1/2)**
>
> We thank the reviewer for their timely engagement during the discussion period and efforts to improve our work. We have now directly linked our additional results in the appendix to our experimental section (Section 5.1). We continue the interesting discussion on the reviewer's points below:
>
> ## Follow-Up to Noisy-TVs and Relevance Functions
>
> Indeed, we acknowledge that Table 5 demonstrates that different relevance functions can lead to different outcomes. Choosing the right relevance function for a particular task or environment is thus an important framework-level hyperparameter.
>
> Our perspective remains that this choice is relatively simple and less heuristic than many other choices commonly made in RL frameworks. We believe the conclusions from Table 5 of Appendix A should be that:
>
> 1. ICM is a good default choice for $\mathcal{F}$ (PGR (Curiosity) with no hyperparameter tuning outperforms the PPO baseline)
>
> 2. ICM failing is a fundamental problem in exploration, endemic to many prediction-error based metrics. This problem is reasonably easy to visually diagnose (inspection of behavior learned by the PPO + Curiosity agent, as documented by Savinov et al. [3]) and quantitatively diagnose (performance of PGR (RND) also suffers).
>
> 3. Choosing other more robust relevance functions such as ECO restores the advantages of PGR, allowing it to outperform the baseline that directly adds an ECO exploration bonus to model-free PPO.
>
> Finally, we buttress our claim that $\mathcal{F}$ can be chosen to inform us apriori of transition tuples that are physically more/less desirable, and promote/penalize generating such transitions through appropriate conditioning. We point out that non-linear MPC-based controllers offer exactly the physically-grounded relevance functions we describe. Concretely, consider the task of quadruped locomotion. Then, the cost functions described in the optimal control problems formulated by Equations 9 or 10 of Corberes et al. [10] offer a concrete way to "score" motion trajectories. Specifically, during online learning of a neural network-based controller, given a particular state (e.g. proprioceptive measurements such as center-of-mass linear/angular velocity/acceleration) and control parameter (e.g. ground reaction forces), we can obtain a "cost" penalization term via a relevance function adopting the form of Equation 9 or 10 of Corberes et al. [10]. This is exactly the scalar conditioning term our diffusion model will accept. We can use a similar prompting strategy as described in Section 4.3 of our main text to densify the trajectories which exhibit the lowest $p\%$ of "cost," for some hyperparameter $p$. This line of reasoning extends to other domains (e.g. grasping/manipulation), where constraints on forces/dynamics are well understood from a model predictive control perspective (e.g. force closures and grasping kinematics in [11]).
>
> Thus, PGR can also condition on relevance functions outside traditional measures of priority in PER literature or exploration bonuses in exploration literature. Again, in the interest of time, we cannot replicate the experimental setups of either [10] or [11], and so ascertaining the utility of PGR under these specific settings is not done. However, we believe it is clear that such formulations are an interesting nascent area of future work that PGR sets the foundation for.
>
>
> [10] Corbères et al. Comparison of predictive controllers for locomotion and balance recovery of quadruped robots. ICRA 2021.
>
> [11] Gold et al. Model predictive interaction control for force closure grasping. CDC 2021.

---

> > ### Author Response · Authors · 2024-11-20
> > **Continuing Discussion with Reviewer 1k48 (2/2)**
> >
> > ## Follow-Up to Comparison to Model-Based Methods
> >
> > We better clarify the difference between " 'densification of the existing data distribution' and 'improving the data distribution' via PGR and Exploration, respectively."
> >
> > We would like to highlight the results in Table 5 of Appendix A and Table 6 of Appendix B, which demonstrate that adding direct exploration bonuses of various kinds, across different tasks and environments, underperforms PGR. Moreover, we believe Table 7 of Appendix B makes a preliminary, albeit tenuous, argument that exploration may be entirely complementary to PGR in certain other tasks and environments. Indeed, these results indicate that there may also be a difference, if only at an empirical level, between PGR and exploration.
> >
> > On a more abstract or ideological level, we strongly believe these two approaches are conceptually different because:
> >
> > 1. Densification can benefit from the generalization capabilities of the generative (diffusion) model, interpolating transitions to hitherto unseen regions of the MDP. On the other hand, exploration-centric approaches require us to explicitly access those transitions during interaction rollouts and enter them into our replay buffer before we can train on them.
> >
> > 2. Densification is directly concerned with improving the learning dynamics of the policy (e.g. via reducing overfitting of the Q functions), whereas exploration is predominantly concerned with completing the data distribution to e.g. better access sparse rewards or identify transitions which can possibly lead to more rewarding behaviors.
> >
> > 3. PGR can choose any relevance function $\mathcal{F}$ for conditioning, for different desired end goals (we concretize this original claim in the future experiment spaces proposed above.) Exploration-based functions are merely one such possible set of instantiations for $\mathcal{F}$.

---

> > > ### Author Response · Authors · 2024-11-25
> > > **Following up on Discussion Period**
> > >
> > > Dear reviewer 1k48,
> > >
> > > Thank you for the constructive feedback and suggestions. We appreciate your engagement in the discussion period; if you feel that we have sufficiently addressed your questions and alleviated your concerns, we would also appreciate if you would consider updating your score.
> > >
> > > Please let us know if there are other additional clarifications we can provide. Thank you once again.

---

> > > > ### Author Response · Authors · 2024-12-02
> > > >
> > > > Dear reviewer 1k48,
> > > >
> > > > We would like to provide a gentle reminder that today is the last day for author-reviewer discussions. If there are any remaining concerns, we would be grateful for an opportunity to address them. We would also appreciate if you could double check that your final score reflects your opinion after engaging with us during this discussion period.
> > > >
> > > > Thank you again for your service.

---

### Author Response · Authors · 2024-11-19
**General Response (1/2)**

We thank the reviewers for their helpful comments on our work. We are glad to see the reviewers believe our problem setting is “an important one to improve the performance of RL methods” (reviewer ir9Q) and that our work in this setting is a “valuable contribution to the RL community” (reviewer Yyu8). Thank you also for finding our approach to be "simple" (reviewers 1k48, Yyu8), "reproducible" (reviewer Yyu8), "novel" (reviewer 8yrp) and "worth integrating with other approaches" (reviewer 1k48). We are especially grateful that the reviewers recognized the importance of our scaling experiments for “enabling more data-efficient training of large-scale RL agents” (reviewer 8yrp) and the resulting insights to be "compelling and valuable" (reviewer 1k48).

We now turn to two common questions amongst the reviewers, before addressing specific comments in individual responses below.

### 1. Choice of Relevance Function $\mathcal{F}$ and the Noisy-TV Problem (Reviewers 1k48, ir9Q, 8yrp)

**Our framework is compatible with a wide variety of relevance functions, such as RND [1] or pseudo-count [2], which have all been demonstrated in prior work to be robust to the noisy-TV problem.**

We have added to the appendix additional results on our pixel-based DMC-100k benchmark, which demonstrate that PGR conditioned on more robust curiosity-based metrics continue to enjoy strong performance. In particular, we look at intrinsic rewards obtained via Random Network Distillation (RND) [1] and pseudo-counts from a context tree switching (CTS) density model [2]. For training details we refer reviewers to Appendix A.1. We reproduce the results here:

|                 | Walker-Walk            | Cheetah-Run            |
|-----------------|------------------------|------------------------|
| DrQ-v2          | 514.11 $\pm$ 81.42     | 489.30 $\pm$ 69.26     |
| SynthER         | 468.53 $\pm$ 28.65     | 465.09 $\pm$ 28.27     |
| PGR (RND)       | **602.10 $\pm$ 43.44** | 512.58 $\pm$ 23.81     |
| PGR (CTS)       | 540.78 $\pm$ 88.27     | 508.17 $\pm$ 74.03     |
| PGR (Curiosity) | 570.99 $\pm$ 41.44     | **529.70 $\pm$ 27.76** |

**But what about more complex environments which actually have noisy TVs? We investigate the setting described in Savinov et al. [3], and simultaneously demonstrate that PGR is also compatible with relevance functions defined via (online) episodic curiosity (ECO).**

In particular, we use the “noise”-randomized versions of the DMLab environment in Savinov et al. [3]. This environment features a maze navigation task, where visual inputs constantly feature uniform random RGB noise in the lower right portion of the image (i.e. a noisy TV). For a visualization of the environment and training details, we refer reviewers to Appendix A.2, and also to Tables S12 and S13 in Section S6 of Savinov et al. [3]. We reproduce the results here:

|                 | Sparse             | Very Sparse        |
|-----------------|--------------------|--------------------|
| PPO             | 7.3 $\pm$ 2.7      | 4.1 $\pm$ 2.4      |
| PPO + Curiosity | 5.6 $\pm$ 1.8      | 2.8 $\pm$ 1.0      |
| PPO + RND       | 8.2 $\pm$ 1.9      | 4.0 $\pm$ 1.1      |
| PPO + ECO       | 16.3 $\pm$ 3.6     | 12.9 $\pm$ 1.9     |
| PGR (Curiosity) | 9.3 $\pm$ 2.0      | 5.7 $\pm$ 2.2      |
| PGR (RND)       | 11.2 $\pm$ 1.0     | 8.0 $\pm$ 1.8      |
| PGR (ECO)       | **21.9 $\pm$ 2.6** | **18.7 $\pm$ 2.1** |

We see that PGR can flexibly condition on relevance functions like episodic curiosity to restore its superior performance in environments with highly stochastic transitions.

**But more crucially, generative replay with the “right” relevance function $\mathcal{F}$ is always better than without.**

Our contribution is not related to effective intrinsic exploration in stochastic environments, or even generally robust conditioning choices for generative replay. We simply seek to make the case that online generative replay is a compelling paradigm for sample-efficient online RL. We rely on the insight that prioritized replay can be elegantly connected to, and easily instantiated via, conditional generation in generative replay. Finally, our mechanistic explanations do not rely on showing that the environment is better explored, which is a common underlying narrative in intrinsic motivation literature and the noisy-TV problem. We only demonstrate the possibility that PGR improves learning dynamics and reduces the prolific problem of early overfitting in Q-learning.

---

> ### Author Response · Authors · 2024-11-19
> **General Response (2/2)**
>
> ### 2. Contextualizing Exploration Bonuses Within the PGR Framework (Reviewers 1k48, Yyu8, ir9Q)
>
> **Our contributions are orthogonal to exploration. We do not make any claims about improved exploration, nor is this a focal point of our problem setting. Nonetheless, we show here that existing work in enhancing exploration can be easily and effectively integrated into the PGR framework.**
>
> We first direct the reviewers to Figure 3b of the main text, which offers initial evidence that PGR goes beyond simple exploration bonuses. In particular, providing either the unconditional replay baseline (SynthER) or the model-free RL baseline (REDQ) with an exploration bonus, via intrinsic curiosity, underperforms PGR. We provide quantitative numbers in Table 6 of Appendix B, along with a repeat of this experiment using RND [1]. We reproduce the results here:
>
> |                     | Quadruped-Walk         | Cheetah-Run            |
> |---------------------|------------------------|------------------------|
> | REDQ                | 496.75 $\pm$ 151.00    | 606.86 $\pm$ 99.77     |
> | SynthER             | 727.01 $\pm$ 86.66     | 729.35 $\pm$ 49.59     |
> | REDQ + Curiosity    | 687.14 $\pm$ 93.12     | 682.64 $\pm$ 52.89     |
> | SynthER + Curiosity | 803.87 $\pm$ 41.52     | 743.39 $\pm$ 47.60     |
> | PGR (Curiosity)     | **927.98 $\pm$ 25.18** | **817.36 $\pm$ 35.93** |
>
>
> For further experimental analysis, we also add two baselines which modify neural network architectural inductive priors to implicitly improve exploration. In particular, we adapt both Bootstrapped-DQN [4] and NoisyNets [5] to our REDQ baseline. For training details and results we refer reviewers to Appendix B (c.f. Table 7). We reproduce the results here:
>
> |                 | Quadruped-Walk         | Cheetah-Run            |
> |-----------------|------------------------|------------------------|
> | NoisyNets       | 688.29 $\pm$ 65.55     | 770.14 $\pm$ 70.53     |
> | Boot-DQN        | 721.49 $\pm$ 39.82     | 754.56 $\pm$ 67.38     |
> | PGR (NoisyNets) | **939.32 $\pm$ 36.74** | 893.67 $\pm$ 43.47     |
> | PGR (Boot-DQN)  | 903.29 $\pm$ 40.50     | **912.65 $\pm$ 47.22** |
> | PGR (Curiosity) | 927.98 $\pm$ 25.18     | 817.36 $\pm$ 35.93     |
>
> **We argue that densification under PGR is empirically orthogonal to and more useful than simply promoting exploration.**
>
> We provide some simple intuition on this conclusion, which also further distinguishes our exploration-agnostic contributions: PGR densifies the relevant subset of _existing_ data to improve learning dynamics, whereas exploration is primarily concerned with improving the distribution of this data in the first place. We do not make this conclusion in the main text, as DMC/OpenAI Gym benchmarks may be too facile for exploration-centric conclusions. Nevertheless, taken together, the comparative experiments above strongly suggest that generative replay in PGR is effective and orthogonal to exploration bonuses.
>
> [1] Burda et al. “Exploration by Random Network Distillation.” arXiv 2018.
>
> [2] Bellemare et al. “Unifying Count-Based Exploration and Intrinsic Motivation.” NeurIPS 2016.
>
> [3] Savinov et al. “Episodic Curiosity Through Reachability.” arXiv 2018.
>
> [4] Osband et al. “Deep Exploration via Bootstrapped DQN.” NeurIPS 2016.
>
> [5] Fortunato et al. “Noisy Networks for Exploration.” ICLR 2018.

---

### Meta-Review · Area_Chair_s4Kk · 2024-12-21

**Metareview:**

This paper uses generative models for samples for experience replay in RL. The paper uses conditional diffusion models, and explores different relevance functions, showing empirical improvements across experiments in online RL.

Reviewers agree that this paper is well-written, and the research problem is important and well-addressed. The method is novel and the results are strong, as well as the insights as to why the method works.

Reviewers had numerous concerns about the relevance function (eg robustness), which the authors addressed convincingly in their rebuttal (and reviewers have increased scores as a result). All reviewers agree that this work is above the acceptance threshold, with most agreeing it is a clear accept. Overall, this is a strong paper which will be a good addition to ICLR.

**Additional Comments On Reviewer Discussion:**

A common question during the original reviews was about the choice of relevance functions. The authors added more results to tackle this and provide further insight, which the reviewers appreciated. The authors added an impressive amount of new results during the rebuttal period to address many of the reviewers' concerns.

Reviewer Yyu8 commented about using other generative models, but I agree with the authors that this comparison is not necessary / can make the messageworse. The authors still provided results with VAEs in the rebuttal. Overall Reviewer Yyu8's review is very short, but they are still positive about the paper. Similarly, Reviewer ir9Q's review is short, but raised some interesting questions that the authors addressed in the rebuttal. Reviewer ir9Q thinks the method is simple, but other reviewers did not agree.

---

### Decision · Program_Chairs · 2025-01-22

Accept (Oral)